# Electronic structure blurring-mediated solid-state $H_2O_2$ electrosynthesis with high productivity

Yuxiang Zhang[1], Jingjing Duan [2], Markus Antonietti [3] & Sheng Chen [1,3] ✉

The development of $H_2O_2$ economy is hampered by the instability of liquid-state bulk $H_2O_2$ solutions ($2H_2O_2 \rightarrow 2H_2O + O_2$; $\Delta G° = -117$ kJ mol$^{-1}$). Comparatively, dispersing $H_2O_2$ molecules in solid-state materials would offer good physical stability with less of handling, leak and exposure risks, but suffers from fabrication schemes irrelevant to commercial applications. Mediated by the concept of electronic structure blurring, here we elaborate one-step electrosynthesis of solid-state $H_2O_2$ with productivity up to 0.943 mol L$^{-1}$ h$^{-1}$. Notably, the as-fabricated solid-state $H_2O_2$ features not only high $H_2O_2$ gravimetric densities ( > 30 wt%) but also good stability for repeated $H_2O_2$ loading/deloading over 100 cycles and shelf life over 160 days. Mechanism study underscores the electronic structure blurring formed at local catalytic environments that contributes to homogenizing charge distributions of H-O and O-O bondings (charge transfer of 0.67 and 0.22 e), and thereby inhibiting the break of these bonds inside $H_2O_2$ molecules. The revelation that "stabilized $H_2O_2$" can be manufactured under industrial conditions offers a path towards a sustainable $H_2O_2$ production.

Conversion of renewable sources (like solar and winds) into electricity is an appealing pathway for developing sustainable society[1]. However, because of the temporal and spatial fluctuations of renewable sources, it is challenging to realize a steady generation of electricity; thus, the storage of electricity in the form of energy carrier is required[2]. In a hydrogen peroxide ($H_2O_2$) economy[2,3], unstable $H_2O_2$ molecule is illustrated as a carrier of renewable electricity, beneficial from its high specific energy density (3.0 MJ$^{-1}$, 60 wt%), hydrogen percentage ($H_2$, 5.9 wt%) and only degradation product of water[2,4]. While the feasibility of $H_2O_2$ economy has been frequently questioned owning to the instability of $H_2O_2$ molecules, which are prone to decompose into $H_2O$ and $O_2$ ($2H_2O_2 \rightarrow 2H_2O + O_2$; $\Delta G° = -117$ kJ mol$^{-1}$) associated with elevated temperatures/pHs[5,6], light illumination[7] and/or trace metal ion impurities (like $Fe^{2+}$, $Cu^{2+}$, and $Mn^{2+}$)[8,9]. The corrosion, spill and leakage hazards of liquid bulk $H_2O_2$ solutions (particularly at high concentrations >30 wt%) create

dangerous runaway situations in the scheme of storage, transport and applications.

Comparatively, the dispersion of $H_2O_2$ molecules in solid-state materials can offer good physical stability with less of handling, leak and exposure risks. This would form a class of "solid-state $H_2O_2$" for sustainable $H_2O_2$ economy, e.g., starting by storing $H_2O_2$ molecules in solid host compounds, followed by transporting to consumption places, which is then released from the storage materials, leaving the primary solid compound for further storage to closes the cycle:

$$H_2O_2 + \text{host materials} \rightleftharpoons H_2O_2 * \text{host materials}$$

To develop such an economy, a prerequisite for solid-state $H_2O_2$ materials is, among many other factors, high $H_2O_2$ gravimetric densities, $H_2O_2$ loading/deloading cycles and long shelf life. The improvement of performances requires each of these parameters to be

[1]Key Laboratory for Soft Chemistry and Functional Materials, School of Chemistry and Chemical Engineering, Nanjing University of Science and Technology, Nanjing, China. [2]School of Energy and Power Engineering, Nanjing University of Science and Technology, Ministry of Education, Nanjing, China. [3]Max Planck Institute of Colloids and Interfaces, Potsdam, Germany. ✉e-mail: sheng.chen@njust.edu.cn

optimized, but increasing one of them without compromising the others is difficult. For example, a vast array of materials can store $H_2O_2$ with high gravimetric densities (>30 wt%), but few of them can sustain loading/deloading cycles because of too strong (chemical bindings, like metal peroxides)[10] or weak interactions (like van der waals forces). The repeated loading/deloading cycles requires reversible bindings between $H_2O_2$ molecules and host materials, while this leads to structure deterioration and consequently short shelf life (i.e., stability-cyclability trade-off).

Actually, researches since decade ago have already proposed the concept of solid-state $H_2O_2$ (i.e., $CaO_2 \cdot 2H_2O_2(s)/H_2O_2$)[11]. Yet in practice, no examples have been reported to satisfy all above criteria. In the preliminary research of this work, we have unexpectedly overcome the stumbling relationship of stability-cyclability trade-off. We have experimentally demonstrated peroxosolvates, a category of solid compounds formed by $H_2O_2$ molecules interacting with metal salts (i.e., $KF \cdot H_2O_2$, $CO(NH_2)_2 \cdot H_2O_2$ and $Na_2CO_3 \cdot 1.5H_2O_2$)[12], which can show not only high $H_2O_2$ gravimetric densities (32.5–36.9 wt%) but also good physical stability for $H_2O_2$ loading/de-loading over 100 cycles and long shelf life over 160 days (Fig. 1g, h; Supplementary Figs. 1–6 and Supplementary Table 1). This result indicates peroxosolvates a very promising candidate for developing $H_2O_2$ economy, despite of their production schemes based on performed $H_2O_2/O_2$ feedstocks (Fig. 1d)[12,13], which are irrelevant to commercial applications because of high cost, complicated produces and/or productivities two orders of magnitude below benchmark industrial anthraquinone process (0.588 mol $L^{-1}$ $h^{-1}$).

As comparison to $H_2O_2/O_2$, atmospheric air is unarguably the most abundant and cost-free source for oxygen-involved chemical reactions[14], although the direct air fixation known to be challenging, and according to Le Chatelier's principle[15], generally leading to compromised reaction rates owning to low $O_2$ level (21%). Recently, we and other groups have addressed this problem by reporting direct air-to-diluted $H_2O_2$ (<10 wt%) conversion at high selectivity by mediating gas diffusion electrodes[16,17], ionic liquid[18] and/or catalyst materials[19,20]. By blending this conversion with host materials, it would be possible to directly produce solid-state $H_2O_2$ from atmospheric air feedstock. Yet, the conversion productivity is still unable to rival anthraquinone route to $H_2O_2$ production, which is primarily due to the inherent instability of $H_2O_2$ molecules. Whenever the key $H_2O_2$ product is in situ generated and accumulated to high concentrations at local conditions, it spontaneously decomposes into $H_2O$ and $O_2$. In line with this hypothesis, preventing $H_2O_2$ decomposition, and thereby stabilizing $H_2O_2$ at local environments, should promote efficient solid-state $H_2O_2$ electrosynthesis.

In this work, we report to stabilize $H_2O_2$ by borrowing the concept of "blurring effect" from image processing research field. Generally, blurring effect is a parasitic process of removing noise and artifacts during image optimization, which exhibits negative role of reducing image quality[21,22]. Here we extend this concept to electrochemistry and describe a stabilization mechanism of electronic structure blurring, which arises from charge redistribution between electrochemically generated $H_2O_2$ and host compounds (e.g., KF). This effect then homogenizes electron density across H–O and O–O bonds, blurring

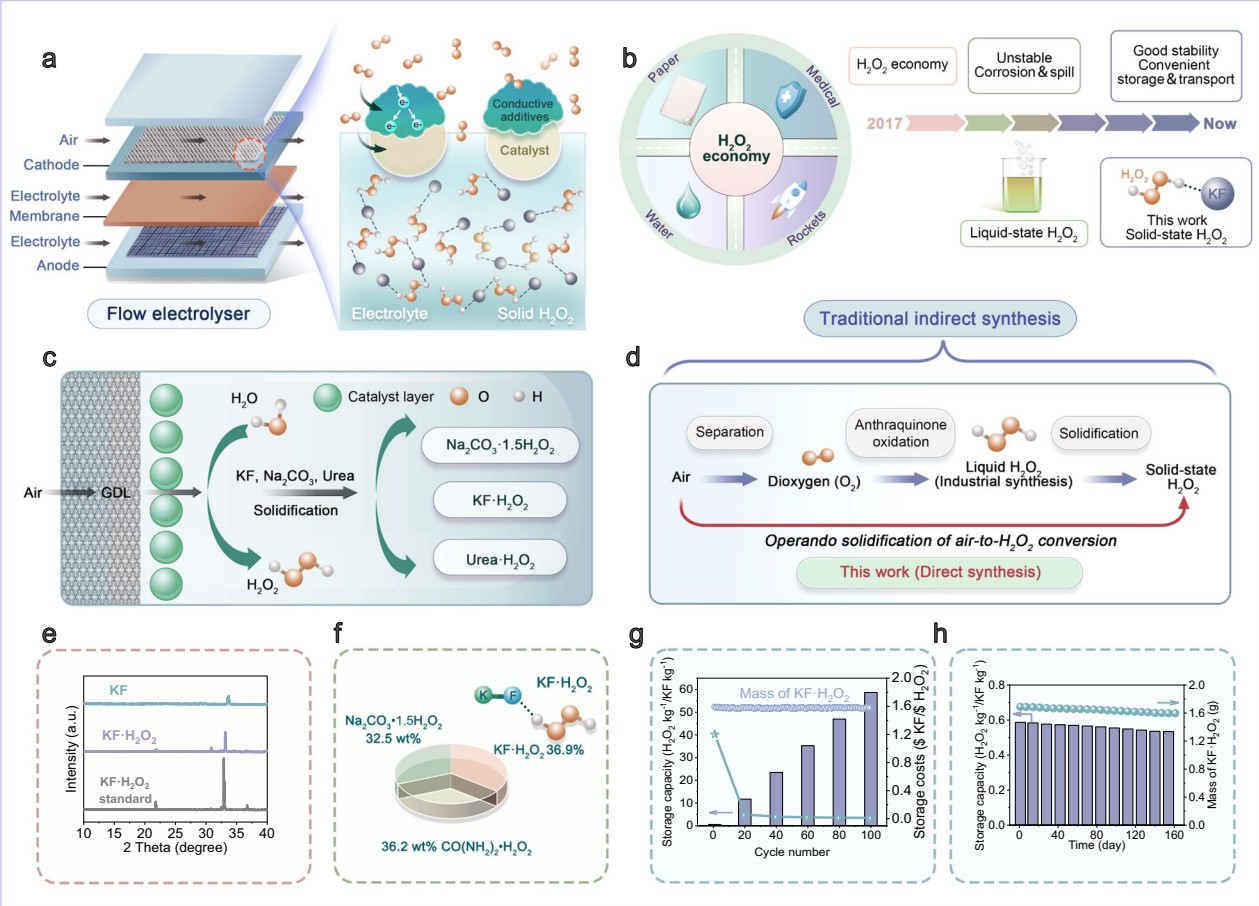

**Fig. 1 | Schematic illustration of $H_2O_2$ production. a** Direct solid-state $H_2O_2$ electrosynthesis in flow cells. **b** Overview of the development for $H_2O_2$ economy. **c** Sketch of solid-state $H_2O_2$ electrosynthesis in the form of peroxosolvates at three-phase boundary. **d** Traditional and the present production diagrams for solid-state $H_2O_2$. **e** XRD patterns of $KF \cdot H_2O_2$ as comparison to KF. **f** Theoretical $H_2O_2$ storage percentages for peroxosolvates like $KF \cdot H_2O_2$, $CO(NH_2)_2 \cdot H_2O_2$ and $Na_2CO_3 \cdot 1.5H_2O_2$. **g, h** Changes in accumulative $H_2O_2$ storage capacity and cost for $KF \cdot H_2O_2$ after 100 cycles or 160 days.

their localized charge polarization to prevent bond cleavage, and promoting solid-state $H_2O_2$ electrosynthesis approaching theoretical limit efficiency (93.3% at the limit current of -350 mA cm$^{-2}$). Experimentally, starting in an air-to-$H_2O_2$ conversion system in a flow-type cell (Fig. 1a), we have explored each possible parameter of local environments (like pHs[23], solvents[24], wettability[25] and electric fields[26]), and unexpectedly observed *operando* interaction formed between solutes (i.e., KF) and local $H_2O_2$ molecules, which enables in situ electrosynthesis of solid-state $H_2O_2$. Notably, $O_2$ from air diffuses through gas diffusion layer, adsorbs and is activated on the catalyst surface, which then couples with protons (H$^+$) generated from $H_2O$ electrolysis to form $H_2O_2$. The *operando* formation of solid-state $H_2O_2$ has been achieved generally through its reaction with KF, urea, and $Na_2CO_3$, leading to the formation of peroxosolvates (Fig. 1c). By circumventing the instability of $H_2O_2$, this method offers a strategy that eliminates the need for the cumbersome and energy-intensive steps of traditional indirect solid $H_2O_2$ synthesis, enabling direct *operando* electrosynthesis of solid-state $H_2O_2$ (Fig. 1b, d).

## Results

### The feasibility of solid-state $H_2O_2$ storage in peroxosolvates

To facilitate $H_2O_2$ sequestration in solid compounds, the first requirement is high gravimetric density (wt.%). Figure 1e, f and Supplementary Fig. 1 present three primary proposed compounds, KF·$H_2O_2$, CO(NH$_2$)$_2$·$H_2O_2$ and $Na_2CO_3$·1.5$H_2O_2$, chemically synthesized via the following equations:

$$KF + H_2O_2 \rightarrow KF \cdot H_2O_2 \qquad (1)$$

$$CO(NH_2)_2 + H_2O_2 \rightarrow CO(NH_2)_2 \cdot H_2O_2 \qquad (2)$$

$$Na_2CO_3 + 1.5H_2O_2 \rightarrow Na_2CO_3 \cdot 1.5H_2O_2 \qquad (3)$$

Experimental evidence for formation of these peroxosolvates has been demonstrated by X-ray diffractions (XRD, Fig. 1e and Supplementary Fig. 2) and Fourier transform infrared spectra (FTIR, Supplementary Fig. 3). Mass ratio calculations show the nominal $H_2O_2$ gravimetric densities of 36.9 wt% for KF·$H_2O_2$, 36.2 wt% for CO(NH$_2$)$_2$·$H_2O_2$ and 32.5 wt% for $Na_2CO_3$·1.5$H_2O_2$, respectively (Fig. 1f). These peroxosolvates can be isolated as stable powder samples under ambient condition owing to intermolecular bondings formed between H atoms (from $H_2O_2$) and F/N/O atoms (from KF/CO(NH$_2$)$_2$/$Na_2CO_3$).

The second requirement for solid-state $H_2O_2$ storage lies in the ability to build up a reasonable loading/deloading cycle with long shelf life of the loaded state. In a typical cycle for KF, the low-temperature heating of KF·$H_2O_2$ simply releases $H_2O_2$, and then KF is left to be used again for further storage that closes the cycle. We explored this process for 100 repetitive storage-release cycles, and KF maintained its storage capacity, with only minor decay in $H_2O_2$ gravimetric densities (<1%, Fig. 1g and Supplementary Table 1). This underlines that side reactions, even with impurities present in the cycle, are rare. Using the 100 cycles lifetime as a base of calculation, the $H_2O_2$ storage capacity of KF increases linearly from 0.587 $H_2O_2$ kg$^{-1}$/KF kg$^{-1}$ to 58.73 $H_2O_2$ kg$^{-1}$/KF kg$^{-1}$. Correspondingly, the cost per unit mass of KF decreases sharply from 1.2 to 0.06 \$ KF/\$ $H_2O_2$ (after 20 cycles) and then gradually to 0.012 \$ KF/\$ $H_2O_2$ (after 100 cycles). We also analyzed the shelf life and found for KF·$H_2O_2$ stored under ambient, standard lab conditions a mass decline of 5.28% within 160 days (-mass decline, Fig. 1h). Similar results have also been recorded for CO(NH$_2$)$_2$·$H_2O_2$ and $Na_2CO_3$·1.5$H_2O_2$ (Supplementary Figs. 5, 6), thus highlighting the potential of using these peroxosolvates for $H_2O_2$ storage.

## Direct solid-state $H_2O_2$ electrosynthesis

A key step for a future $H_2O_2$ energy cycle lies in the efficient electrochemical synthesis of such peroxosolvates, ideally from stranded green electricity and atmospheric air feedstock. For illustration, we have synthesized a metal-organic framework-derived catalyst for this purpose. The synthesis started by the coordinative assembly of Zn nodes and organic ligand (2-methylimidazole), followed by low-temperature calcination at 350 °C (the product is denoted as ZIF-350; Fig. 2a, b and Supplementary Figs. 7–15). Firstly, the catalytic activities of air-to-$H_2O_2$ solidification have been evaluated in KF electrolyte (Fig. 2c), showing oxygen reduction with electron transferred numbers of 2.1–2.5 in a rotating ring-disk electrode (RRDE) system, closely aligning with the theoretical two-electron transfer pathway (Supplementary Fig. 20). Quantitatively, the $H_2O_2$ Faradaic efficiencies (FEs) of ZIF-350 consistently exceeded 90% across the entire range of applied current densities, the values being 99.5–93.3% at 50–350 mA cm$^{-2}$, respectively (Fig. 2d, e and Supplementary Figs. 21–24). Even at the theoretical limiting current density of air-to-$H_2O_2$ solidification (-350 mA cm$^{-2}$), the FE reaches 93.3%, together with the high $H_2O_2$ yield rate of 30.64 mol g$_{cat}^{-1}$ h$^{-1}$. The great activities for ZIF-350 have been further verified by high Faradaic efficiencies at low $O_2$ concentrations (5, 10 and 15% $O_2$; Fig. 2f and Supplementary Figs. 25–27), in addition to a high durability for 80 h with minimal fluctuation in Faradaic efficiencies at 200 mA cm$^{-2}$ (Fig. 2h and Supplementary Figs. 28, 29).

Interestingly, without KF, the Faradic efficiencies for ZIF-350 substantially decline to 84.9% at 350 mA cm$^{-2}$ in $K_2SO_4$ electrolyte (Supplementary Fig. 30). We attribute that to the stabilization of $H_2O_2$ by KF in the form of peroxosolvates already from the solution phase. This is also reflected in the analysis of $H_2O_2$ self-decomposition in $K_2SO_4$ electrolyte (Fig. 2g), where the overall $H_2O_2$ Faradaic efficiencies continuously decrease from 93.35 to 67.52% ($H_2O_2$ concentrations from 6.96 to 25.26 mmol) after prolonged electrolysis from 1 to 10 h. This is to be compared with data in KF electrolyte, where overall $H_2O_2$ Faradaic efficiencies only decline from 98.7 to 93.6%, while the cumulative $H_2O_2$ concentration nearly linearly grows from 7.36 to 34.9 mmol. The stabilization of $H_2O_2$ is verified by analyzing the further $H_2O_2$ reduction reaction to $H_2O$, where the activities of $H_2O_2$ reduction in KF are significantly lower when compared to that in $K_2SO_4$ (Supplementary Fig. 31). All these experiments indicate additional advantages of KF solute for stabilizing $H_2O_2$ during electrosynthesis. Notably, the stabilizing mechanism observed with KF can be readily extended to other systems such as urea and $Na_2CO_3$, demonstrating the generality of this approach (Supplementary Fig. 32).

## Mechanism study

The interaction between KF and $H_2O_2$ has been monitored by *operando* Raman spectra in a flow cell at the current density of 200 mA cm$^{-2}$ in 1.0 M KF electrolyte, using an excitation wavelength of 750 nm (Fig. 3a). Starting with pristine KF electrolyte at 0 h, only weak Raman signal could be observed, due to the small Raman scattering cross-section of ionic KF compound. After 2-h electrolysis, a Raman vibration has emerged at 858 cm$^{-1}$, whose peak intensity increases with elongated electrolysis at 4 h (Fig. 3a and Supplementary Fig. 33), leading to a sharp spectral band corresponding to the O–O stretching vibration in $H_2O_2$[20]. This result indicates continuous $H_2O_2$ accumulation in the electrode during electrolysis. Notably, the Raman band in the spectrum is shifted by 15 cm$^{-1}$ compared to commercial $H_2O_2$ (873 cm$^{-1}$). This shift may be attributed to the interaction between KF and $H_2O_2$ produced electrochemically in the electrolyte. This result is consistent to our density functional theory (DFT) calculations revealing a stronger adsorption of KF*$H_2O_2$ at the electrode surface compared to KF*$H_2O$ (adsorption energy: −0.90 vs. −0.50 eV, Fig. 3b), which leads to blur the charge distribution of H–O and O–O bonds inside $H_2O_2$ molecules (charge transfer of 0.67 and 0.22 e) and thereby inhibiting

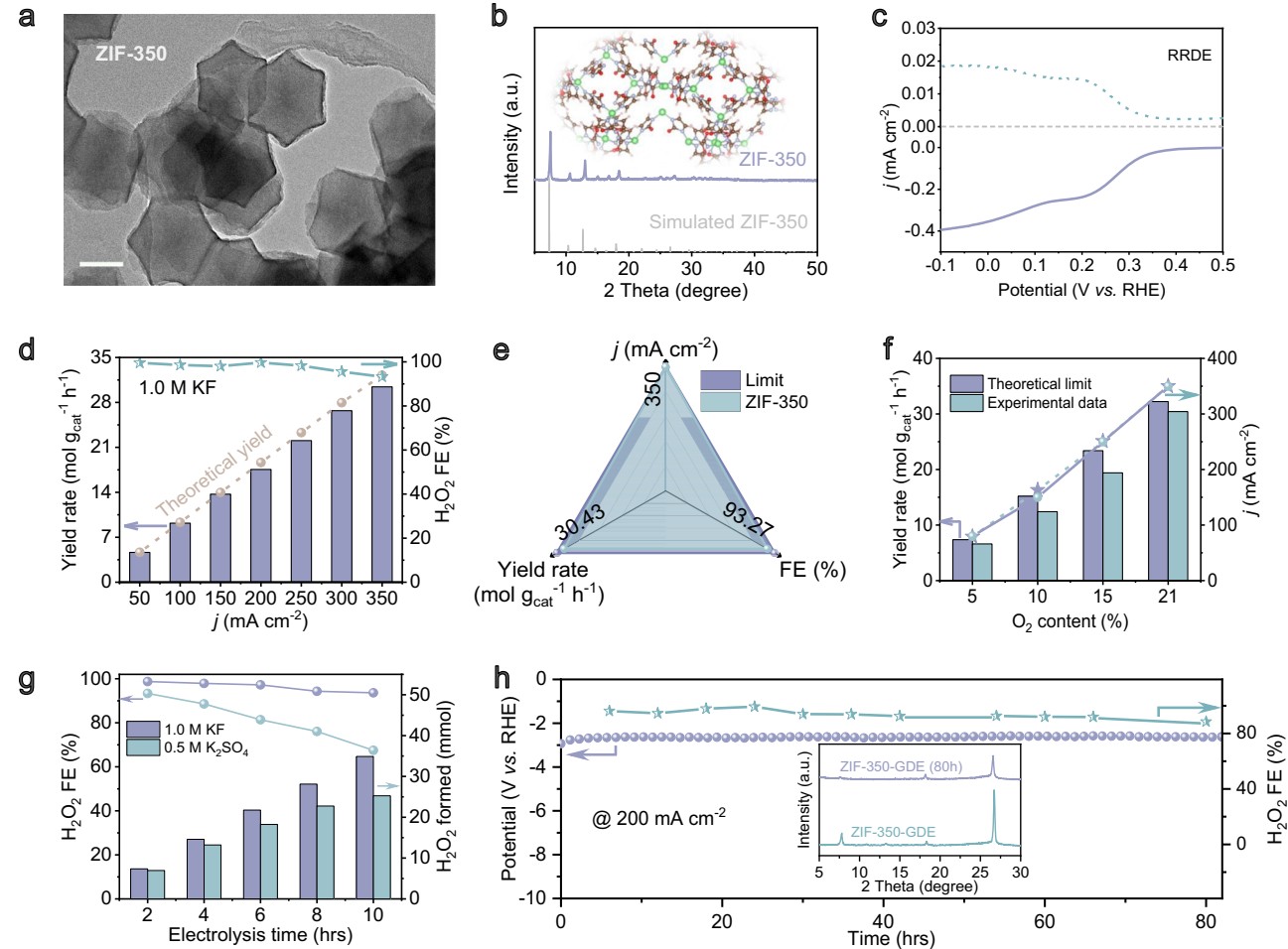

**Fig. 2 | Direct solid-state $H_2O_2$ electrosynthesis. a** High-resolution transmission electron microscopy (HR-TEM) image of ZIF-350 catalyst (scale bar: 200 nm). **b** X-ray diffraction (XRD) pattern and structure illustration of ZIF-350 catalyst. **c** RRDE linear sweep voltammetry (LSV) of ZIF-350 catalyst recorded at a rotation rate of 1600 rpm, along with the detected $H_2O_2$ currents on a Pt ring electrode at a fixed potential of 1.2 V vs. RHE. **d, e** Faradaic efficiency (FE) and yield rates in flow cells. **f** The current densities at low $O_2$-content environments in flow cells. **g** Cumulative $H_2O_2$ concentrations in different electrolytes in flow cells. **h** The stability test at 200 mA cm$^{-2}$ in flow cells, with the inset showing the XRD patterns of catalyst before and after test.

the break of these bonds. To further validate the stabilizing role of electronic structure blurring on $H_2O_2$ molecule, we have calculated bond energies for isolated $H_2O_2$ and $H_2O_2$ in peroxosolvates (i.e., $KF \cdot H_2O_2$, $CO(NH_2)_2 \cdot H_2O_2$ and $Na_2CO_3 \cdot 1.5H_2O_2$, Supplementary Fig. 34)[27]. Generally, the instability of $H_2O_2$ molecule originates from nonuniform charge distributions between different bonds, with a bonding energy difference of 1.07 eV between O–H and O–O bonds. Interestingly, both bond energies of O–H and O–O in peroxosolvate have been reduced, attributable to bond elongation (i.e., O–H···N, O–H···O and O–H···F, Supplementary Fig. 35). The difference between the O–H and O–O bond energies in peroxosolvates has also been reduced to 0.90 eV for $KF \cdot H_2O_2$, 0.91 eV for $CO(NH_2)_2 \cdot H_2O_2$ and 0.92 eV for $Na_2CO_3 \cdot 1.5H_2O_2$, respectively (Supplementary Fig. 34d). Accordingly, we consider the electronic structure blurring between O–H and O–O bonds that enables high stability of $H_2O_2$ molecule. Further molecular dynamics (MD) simulations provide insight into the electronic structure blurring (Supplementary Fig. 36)[28]. It demonstrates that during energy minimization, $H_2O_2$ molecules undergo directed migration towards KF species. This spontaneous reconfiguration, driven by favorable electrostatic interactions and hydrogen bonding, leads to a significant reduction in the total system energy. Our DFT calculations suggest potential trends in the catalytic mechanism rather than definitive conclusions, given the inherent challenges in accurately modeling the exact material structure and its

dynamic evolution under realistic catalytic conditions. We have provided the computational models as Supplementary Data 1, 2, and 3.

With such a stabilization mechanism, KF concentration should played a vital role in the electrochemical reaction (Supplementary Fig. 37). In a diluted 0.5 M KF electrolyte, the system shows Faradaic efficiencies of 99.32–96.31% (yield rates of 4.62–17.95 mol g$_{cat}^{-1}$ h$^{-1}$) in the range of 50–200 mA cm$^{-2}$, which however decline to 85.68% (yield rate: 28.14 mol g$_{cat}^{-1}$ h$^{-1}$) at elevated current densities at 350 mA cm$^{-2}$ (Fig. 3c and Supplementary Fig. 38). After these experiments, we elevated KF concentration, and found that the $H_2O_2$ Faradaic efficiencies increased again, achieving 93.27% for 1.0 M, 91.82% for 2.0 M and 90.35% for 4.0 M KF at 350 mA cm$^{-2}$, respectively. These findings underscore the stabilizing role of electronic structure blurring in regulating $H_2O_2$ electrosynthesis. Further, the electronic structure blurring is not exclusive to KF. Our additional tests using NaF and $K_2CO_3$ as electrolytes were conducted in the same condition. For NaF electrolyte, the $H_2O_2$ Faradaic efficiencies (FEs) of ZIF-350 consistently exceeded 90% across the entire range of applied current densities, the values being 97.5–90.8% at 50–350 mA cm$^{-2}$, respectively (Supplementary Fig. 39a). Even at the theoretical limiting current density of air-to-$H_2O_2$ solidification (-350 mA cm$^{-2}$), the FE reaches 90.8%, together with the high $H_2O_2$ yield rate of 29.62 mol g$_{cat}^{-1}$ h$^{-1}$. On the other hand, for $K_2CO_3$ electrolyte, the $H_2O_2$ Faradaic efficiencies (FEs) of ZIF-350 is 97.9–87.8% at 50–350 mA cm$^{-2}$ (Supplementary Fig. 39b), and

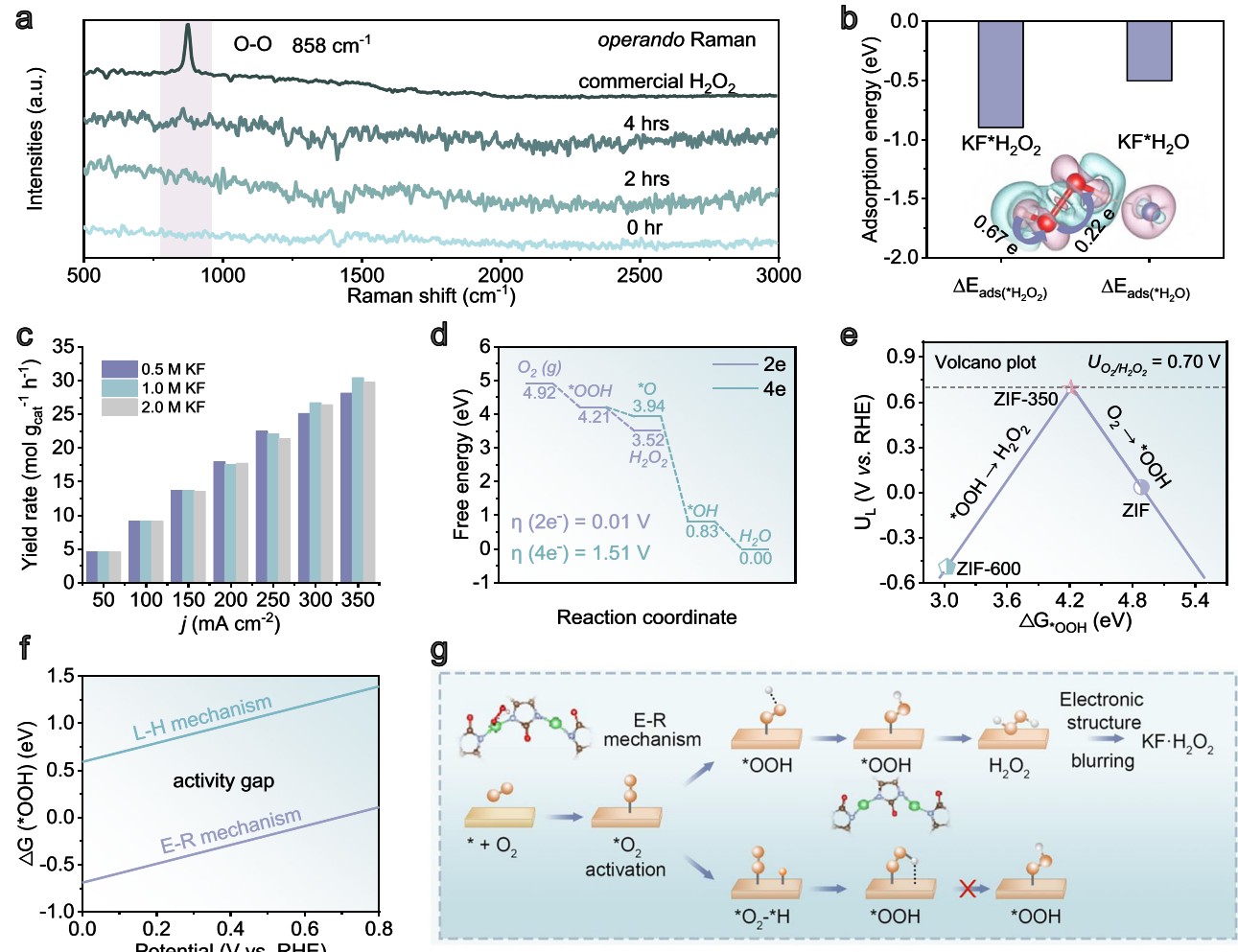

**Fig. 3 | Mechanistic investigation. a** The *operando* Raman spectra for air-to-H₂O₂ conversion. **b** Adsorption energy of KF with H₂O₂ and H₂O (inset shows the charge transport inside KF*H₂O₂, the red, white and purple atoms denote oxygen, hydrogen, and fluorine, respectively). **c** H₂O₂ yield rates in different KF concentrations in flow cells. **d** Free-energy diagrams for 2e- and 4e-ORR pathways. **e** The theoretical volcano curve for 2e-ORR. **f** Kinetic barriers for *O₂ → *OOH *via* Eley-Rideal (E-R) and Langmuir-Hinshelwood (L-H) mechanisms. **g** Schematic air-to-H₂O₂ solidification pathway.

reaching 87.8% and 28.64 mol $g_{cat}^{-1}$ $h^{-1}$ at ~350 mA cm⁻². These results suggest that electrolytes capable of interacting with H₂O₂ can also introduce electronic structure blurring.

We have also modeled the role of the ZIF-350 catalyst to realize such high air-to-H₂O₂ solidification efficiencies (Fig. 3d). The theoretical structure of ZIF-350 has been constructed with cluster model (Supplementary Figs. 40–42) according to experimental characterizations (Fig. 2b and Supplementary Figs. 11–15). On the surface of ZIF-350 catalyst, 2e-ORR pathway occurs with a low energy barrier of 0.01 V thorough two consecutive proton-coupled electron transfer (PCET) hydrogenation steps of O₂ → *OOH → H₂O₂, while the 4e-ORR pathway with a much higher energy barrier of 1.51 V thorough four consecutive steps of O₂ → *OOH → *O → *OH → H₂O. The above result is consistent with the volcano plot (Fig. 3e), where on the left-hand side is ZIF counterpart (synthesized without calcinations; ΔG*OOH = 4.88 eV) with weak adsorption to *OOH intermediate, while on the right-hand side is ZIF-600 counterpart (synthesized by calcinations at 600 °C; ΔG*OOH = 3.03 eV) with strong adsorption to *OOH intermediate. For the ZIF-350, the ΔG*OOH value (4.21 eV) is close to the peak of the volcano (ΔG*OOH of 4.22 eV and limiting potential of 0.7 V)[29], thus delivering the highest 2e-ORR selectivity (Supplementary Fig. 43).

Finally, we have elucidated the origin of the proton for the key O₂ → *OOH step in 2e-ORR along two mechanisms (Supplementary Fig. 44): (i) it could directly stem from the bulk phase of aqueous electrolytes via Eley-Rideal mechanism[30]; (ii) it comes from the adjacent adsorbed *H along a Langmuir-Hinshelwood mechanism[31]. As illustrated in Fig. 3f, ZIF-350 shows a preference for the Eley-Rideal mechanism, exhibiting a free energy change of −0.69 eV, which is significantly lower as compared to 0.59 eV for the Langmuir–Hinshelwood counterpart.

To combine all above information, a scheme is illustrated for the solidification of air-to-H₂O₂ solidification (Fig. 3g), starting by O₂ activation through spontaneous adsorption at zinc sites, followed by the first electron-coupled-proton transfer to produce *OOH intermediate with proton sourced from Eley−Rideal mechanism. Subsequently, the *OOH intermediates take up the second couple of electron and proton to produce *H₂O₂, rapidly desorb from the surface of electrode. In the process of H₂O₂ removal, it simultaneously interacts with KF to form stable peroxosolvate (KF·H₂O₂) pairs, which is stabilized by electronic structure blurring at local conditions, and finally zinc sites are left again on the surface of catalysts that closes this catalytic cycle.

## Scale-up production

We also demonstrated the feasibility of the present catalytic system for a scaled-up production, increasing the electrode area from 1 to 16 cm² by using a flow-cell stack. Firstly, following the conventional configuration, we have built the flow-cell stack comprised of four flow cells by connecting both cathode and anode electrolytes in tandem and

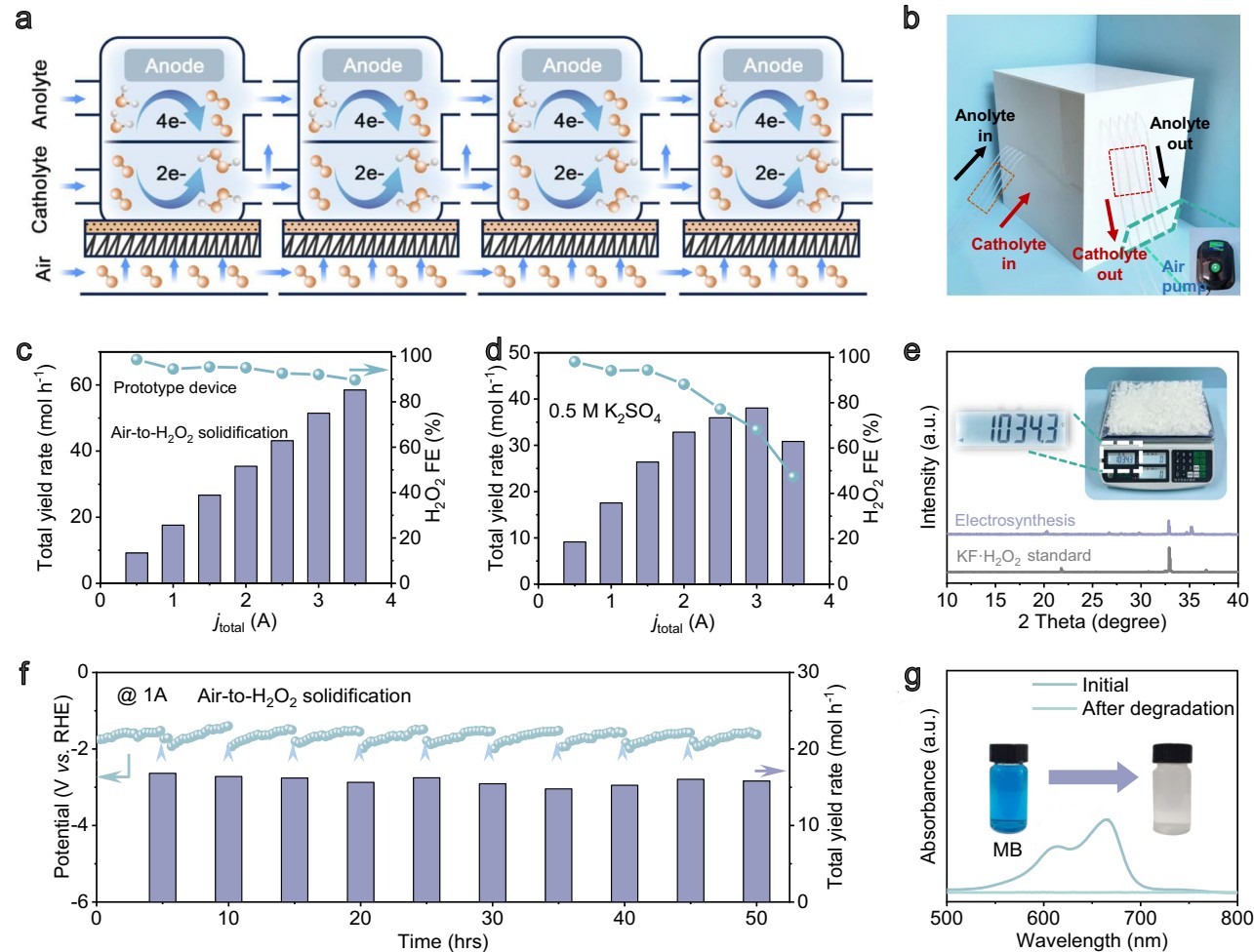

**Fig. 4 | Scale-up production. a** Schematic configuration of our developed tandem/parallel prototype device. **b** The optical picture of the prototype device. **c** The $H_2O_2$ yield rates under ampere-level currents in KF electrolyte. **d** The 2e-ORR performance under ampere-level currents in $K_2SO_4$. **e** XRD patterns of electro-synthesized KF·$H_2O_2$ as comparison to commercial counterpart, with the inset showing the production of kilogram-scale KF·$H_2O_2$. **f** The stability test at 1 A for 50 h. The arrows represent electrolyte refresh. **g** Potential application of KF·$H_2O_2$ for the degradation of methylene blue pollutants.

circuits in parallel (Supplementary Fig. 45). However, here the system achieved only lower activities, with the Faradaic efficiencies of 91.16% at 0.5 A, 92.91% at the applied current of 1.0 A, 73.63% at 1.5 A, and 31.56% at 2 A (Supplementary Fig. 46). This phenomenon is attributable to the instability of $H_2O_2$ prone to self-decomposition or further reduction reaction in the flow-cell stack as discussed above.

Consequently, we have modified the flow-cell stack by keeping the anode electrolyte in tandem, while changed the cathode electrolyte and circuits in parallel (Fig. 4a). The overall prototype device is present in the form of a cuboid in the size of 50 cm × 40 cm × 50 cm consisting of an air pump for feeding air, a peristaltic pump for circulating KF electrolyte and a flow-cell stack equipped with ZIF-350 catalyst (Fig. 4b). Under these conditions, the 2e-ORR activity has been quantified through a chronoamperometric tests at different applied currents (Fig. 4c and Supplementary Fig. 47). The activities of the device show only a minor decay in the current range of 0.5–2.0 A, demonstrating the Faradaic efficiencies of 98.6% at 0.5 A, 94.5% at 1.0 A, 95.4% at 1.5 A, and 95.0% at 2 A, respectively (Fig. 4c). Analogously, the corresponding $H_2O_2$ yield rates are 9.19, 17.61, 26.67 and 35.42 mol h$^{-1}$ at applied currents from 0.5, 1.0, 1.5 and 2.0 A, respectively. Even by further elevating the current to 3.5 A, the prototype device can still maintain high Faradaic efficiency of 89.6% and $H_2O_2$ yield rate of 58.47 mol h$^{-1}$ with the KF electrolyte. For reference, we also analyzed the standard $K_2SO_4$ electrolyte under the same conditions, and significant worse behavior could be

noted (Faradaic efficiency of 47.27% and yield rate of 30.84 mol h$^{-1}$, Fig. 4d). In addition, our device can achieve a higher $H_2O_2$ productivity as comparison to benchmark industrial anthraquinone method[32], indicating that it is indeed a promising candidate for large-scale electrosynthesis. As shown by comparisons with existing literature, our device demonstrates competitive performance relative to a larger share of documented studies when considering air-to-$H_2O_2$ conversion (Supplementary Table 9).

Beside great activities, the potential for industrial production of our flow-cell stack has been further verified by additional tests. For example, the device has demonstrated excellent stability for 50 hrs at ampere-level current, showing seldom decline in yield rates (Fig. 4f). The as-produced KF·$H_2O_2$ in electrolyte was collected, concentrated and dried through evaporation, which can achieve up to 1.0343 kg with purity comparable to commercial counterpart (Fig. 4e). Based on the electrochemical performances of the prototype device, industrial-scale technoeconomic analysis (TEA) has been conducted that shows the capital cost of $H_2O_2$ much smaller than industrial anthraquinone oxidation method ($0.539 vs. 1.5 kg$^{-1}$, Supplementary Fig. 48). Finally, additional application of KF·$H_2O_2$ was demonstrated for direct oxidative degradation of a model dye (Fig. 4g), where a portion of KF·$H_2O_2$ was mixed with an aqueous methylene blue (MB) solution (20 mg mL$^{-1}$). This has triggered vigorous bubbling and a noticeable fading of the solution color as revealed by the optical images and UV–vis spectra.

## Outlook

In recent literature, there is a surge of discussion on the potential of $H_2$[33], $NH_3$[34], and $CH_3OH$ economies and cycles[35], which is based on the fact that these small molecules bearing high specific energy densities, can be made from abundant sources using green electricity, offering well-established production processes and most importantly, showing a chemically stable nature under ambient condition, in spite of their energy content. In this work, we have tried to demonstrate that the chemically rather unstable $H_2O_2$ molecule can be tamed upon solidification with peroxosolvates. To our opinion, this holds great promise for implementing at least in industrial environments a "$H_2O_2$ economy". We illustrate here significant air-to-$H_2O_2$ solidification activities in both single flow-type cell and prototype stacks, while the electronic structure blurring stabilized *operando* preparation process for achieving high concentration storage behavior. This work not only advances the field of large-scale $H_2O_2$ electrosynthesis and applications, but also indicates broader implications for solidifying other unstable molecules (such as HClO and $N_2H_4$)[2] for next-generation energy carriers.

## Methods

### Chemicals

Zinc nitrate hexahydrate ($Zn(NO_3)_2 \cdot 6H_2O$, 99%), Potassium fluoride (KF, 99%), Potassium sulfate ($K_2SO_4$, 99%), Nafion dispersion (5%wt in water and 1-propanol), 2-methylimidazole (98%), were purchased from Alfa Aesar. Methanol and ethanol were obtained from Sinopharm Chemical. Nafion 117 membrane was purchased from Dupont.

### Synthesis of ZIF-350 catalyst

2-methylimidazole (2.464 g) and Zn $(NO_3)_2 \cdot 6H_2O$ (2.232 g) were dissolved in 60 mL and 30 mL of methanol, respectively. Next, these solutions were mixed with stirring for 1 h. The as-obtained ZIF product was washed, centrifuged and dried under vacuum. Finally, the synthesized ZIF sample was placed in a tube furnace to pyrolyze at 350 °C for 2 h (under air atmosphere) with the heating rate of 1 °C min$^{-1}$ (Supplementary Fig. 7).

### Material characterizations

The morphological and structural properties of the materials were investigated using a suite of characterization techniques. A JEOL 7800 F field emission scanning electron microscope (FESEM) was employed for scanning electron microscopy (SEM) imaging. Microstructural analysis was performed via transmission electron microscopy (TEM) on an aberration-corrected FEI Titan 80-300 instrument operating at 300 kV. Crystal phase identification was carried out by X-ray diffraction (XRD) on a Smartlab 9 kW diffractometer (40 kV, 40 mA) using Cu Kα radiation (λ = 1.5418 Å)[36]. Surface chemical states were analyzed by X-ray photoelectron spectroscopy (XPS) using a Thermo ESCALAB 250XI spectrometer with an Al Kα source (1486.6 eV), scanning a binding energy range of 0−1350 eV. Functional groups were characterized by Fourier transform infrared (FT-IR) spectroscopy on a Thermo Fisher NICOLET IS 10 spectrometers. Furthermore, local electronic structure and atomic coordination environment were probed by X-ray absorption near-edge structure (XANES) and extended X-ray absorption fine structure (EXAFS) measurements at the Shanghai Synchrotron Radiation Facility.

### Rotating ring disk electrode (RRDE) system

The rotating ring-disk electrode (RRDE) measurements were conducted using a commercial setup (Pine Research Instrumentation, USA) featuring a glassy carbon disk and a platinum ring. To prepare the working electrode, 5 mg of catalyst was dispersed in a mixture of 1 mL isopropanol/deionized water (3:1 v/v) and 40 µL Nafion solution. After one hour of sonication to form a homogeneous ink, 10 µL of the dispersion was drop-cast onto the glassy carbon disk (area: 0.2475 cm²) and dried to form a uniform catalyst layer[37]. The electrochemical cell was assembled using the catalyst-loaded RRDE as the working electrode, a carbon rod as the counter electrode, and a saturated Ag/AgCl electrode as the reference. ORR performance was evaluated in $O_2$ or air-saturated 0.5 M $K_2SO_4$ electrolyte via linear sweep voltammetry (LSV) at 1600 rpm. During LSV, the Pt ring potential was held at +1.2 V (vs. RHE) to monitor the oxidation of peroxide species.

The hydrogen peroxide selectivity (%) and electrons transferred numbers (n) were derived from the disk ($I_d$) and ring current ($I_r$) currents according to the following equations:

$$H_2O_2 \, selectivity \, (\%) = 200 \times \frac{I_r/N_c}{|I_d| + I_r/N_c} \tag{4}$$

$$n = 4 \times \frac{|I_d|}{|I_d| + I_r/N_c} \tag{5}$$

### Single flow cell assembly

Firstly, the working electrode was prepared by solution casting method. In detail, 5 mg of catalysts and 50 µL of Nafion solution (5 wt%) were dispersed in 950 µL of isopropanol by ultrasonication for 1 hr to form a uniform catalyst ink. The ink was then drop-cast onto carbon fiber paper (28BC) as a cathode using a micropipette. The catalyst loading of 0.2 mg cm$^{-2}$ was determined by weighing the mass of carbon fiber paper before and after spraying, which was dried under ambient conditions to form gas diffusion electrolyte (Supplementary Table 10). The electrocatalytic test were operated on a CHI 760E electrochemical workstation connected with a CHI 680 C high current amplifier. The catalyst-based GDE, Ag/AgCl (saturated KCl) and titanium-based metal oxide coated electrode (DSA, $IrO_2$ coating) were used as working, reference, and counter electrodes, respectively. Cathode and anode compartments were separated by proton exchange membrane (PEM, Nafion 117). In addition, different concentrations of KF and 0.5 M $K_2SO_4$ aqueous solution were used as cathode and anode electrolytes, respectively. The air flow rate used at each current density for the flow cell tests is 20 sccm.

### Prototype flow-cell stack assembly

The device was assembled into a standalone box (size: 50 cm × 40 cm × 50 cm) composed of battery power, gas pump, two peristaltic pumps, four electrolysis cells and two flasks for holding electrolytes. The electrocatalytic performances of this prototype device were tested in natural air. The solidified $H_2O_2$ product from the prototype unit was obtained by evaporation and concentration. Finally, certain amount of the product was mixed in methylene blue with a concentration of 20 ppm to explore its potential for dye degradation experiments.

### Product analyses

The $H_2O_2$ concentrations were quantified by UV–Vis spectroscopy method[38]. In brief, solution A was prepared by mixing 0.4 M of KI, 0.06 M of NaOH, and 10$^{-4}$ M of ammonium molybdate. Solution B was prepared by using a 0.1 M of KHP aqueous solution. Next, equal volumes of solutions A and B were mixed, where a certain amount of $H_2O_2$-containing solution was added. The UV-vis absorbance of the resulted mixture was measured at the wavelength of 351 nm. The $H_2O_2$ yield rate (mol g$_{cat}^{-1}$ h$^{-1}$) and Faradaic efficiency (FE) were calculated according to the following equation:

$$Yield \, rate = \frac{C_{H_2O_2} V_{electrolyte}}{t \, m_{cat}} \tag{6}$$

$$Yield \, rate = \frac{C_{H_2O_2} V_{electrolyte}}{t \, m_{cat}} \tag{7}$$

$$FE\,(\%) = \frac{2FC_{H_2O_2}V_{electrolyte}}{34It} \times 100\% \qquad (8)$$

Where $C_{H_2O_2}$ is $H_2O_2$ concentration (mg L$^{-1}$), $V_{electrolyte}$ is the volume of cathodic electrolyte (L), $F$ is the Faradaic constant (96485 C mol$^{-1}$), 34 is the molar mass of $H_2O_2$ (g mol$^{-1}$), $t$ is reaction duration, $I$ is the applied steady current (A) and $m_{cat}$ is catalyst mass loading (mg cm$^{-2}$).

## Data availability

The datasets generated and analysed during the present study are included in the paper and supplementary information. Source data are provided with this paper.

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

## Acknowledgements

This work was supported by Jiangsu Natural Science Foundation (Grant No. BK 20230097), National Natural Science Foundation (Grant No. 92163124 and 52376193) and the BL20U1, BL17B, and BL14W1 beamline at the Shanghai Synchrotron Radiation Facility.

## Author contributions

S.C. supervised the project and designed the experiments. Y.X.Z. and J.J.D. performed experiments and DFT calculations. S.C., M.A., Y.X.Z., and J.J.D. discussed the results for paper preparation.

## Funding

## Competing interests

Sheng Chen has filed Chinese provisional patent applications (No. 202411655310.7) based on this work. Other authors declare no competing interests.
