## [Transparent Peer Review file · Nature Communications]

Electronic structure blurring-mediated solid-state H₂O₂ electrosynthesis with high productivity

Corresponding Author: Professor sheng Chen

Version 0:

Reviewer comments:

Reviewer #1

(Remarks to the Author)

This paper presents a solid-state electrosynthesis approach for H₂O₂ production, demonstrating a productivity that exceeds that of the conventional industrial process. Traditional H₂O₂ synthesis requires strategies to suppress its decomposition; in this study, the authors introduce peroxosolvates to mitigate H₂O₂ degradation, achieving a productivity of 0.943 mol L⁻¹ h⁻¹, which surpasses that of the benchmark anthraquinone method. The work is significant as it proposes a novel electrochemical system design for efficient peroxide production. I recommend this manuscript for publication with minor revisions. Specific comments are as follows:

Comments

1. The overall image quality is not satisfactory. It is necessary to enhance the resolution of the figures to improve readability.
2. In many cases, the figure descriptions are either missing, insufficient, or inaccurate

For example:

- There is no description provided for Figures 1a and 1b.
- Additional and more specific explanations are needed for the supplementary data.
- In Figure 1g, it is unclear why the storage capacity improves with increasing cycle number; this requires further clarification.
- The purpose of Figure 1f is not clearly explained.
- For Figures 1c and 1d, the schemes are not clearly described, making it difficult to understand what they represent.
- The explanation for Figure/Table 1 is insufficient.
- While the storage capacity appears to improve with an increasing number of cycles, this trend is not addressed in the text and should be discussed.
- There is no explanation provided for Figures 21 through 23.
- In Figure 40, the content of the illustration seems inconsistent with its description.

Overall, many parts of the figures and their corresponding explanations require revision and clarification.

3. Additional proofreading is necessary to correct typographical errors and verify the accuracy of the content. For example, in line 71, it should be confirmed whether "H₂O" is the correct term in the phrase "direct air-to-diluted H₂O (<10 wt%)". Also, in line 86, it is unclear whether the reference to "Fig. 1c and d" is accurate. A more thorough review of such details is recommended.

4. The term "blurring effect" needs to be explained in more detail and with greater clarity regarding its meaning and implications. If possible, including a simulation that calculates and compares the bond dissociation energies of H–H and O–H would further strengthen the explanation.

5. In line 104, it is unclear why H is considered to be bonded with FNO. When explaining this using FTIR data, it would be helpful to include the specific peak assignment that supports this interpretation.

6. In Figure 2f, the experimental peak data for the 10% condition shows a higher current density; however, the corresponding yield appears to be lower. This discrepancy should be addressed or clarified.

7. In line 149, the Raman peak at 0 hours is not clearly visible. Similarly, the peak at 4 hours mentioned in line 151 also appears to be indistinct and should be re-examined.

8. The common characteristics of the additives used in this study—such as KF, Na₂CO₃, and urea—are not clearly

explained. It would be helpful to clarify whether the blurring effect would still occur if alternative salts such as NaF were used instead of KF, or K_2CO_3 instead of Na_2CO_3 . Furthermore, the specific conditions under which the blurring effect occurs should be described in more detail, ideally in relation to the physicochemical properties of the additives.

9. In Figure 37, it is unclear whether the illustrations of the Er and Ih models are accurate. This should be carefully reviewed for correctness.

Reviewer #2

(Remarks to the Author)

The author describes the use of the strong hydrogen-bonding interaction between KF and H_2O_2 to stabilize the H_2O_2 in a solid state KF- H_2O_2 with H_2O_2 concentration $>30\text{wt}\%$. While achieving a high H_2O_2 concentration and the enhanced stability are valuable and important metrics for practical applications, the underlying concept of utilizing hydrogen bond to stabilize H_2O_2 to form solid-state crystalline peroxosolvates has already been extensively explored, in particular, KF has been identified to be one of the most stable form of peroxosolvate hosts. Accordingly, I have the following major concerns regarding the novelty, experimental validation of the proposed mechanism, and the practical applicability of the proposed process:

1. The concept of crystalline peroxosolvates has been widely adopted in the industrial process, where the Na_2CO_3 - H_2O_2 and Urea- H_2O_2 have been the widely adopted options given their less toxic and less hazardous nature. By contrast, the use of KF in the process may raise environmental and safety concerns that could limit the practical applicability of this process. It is suggested that the author provides a more in depth discussion of the sustainability considerations associated with KF. Or please identify a specific use case in which H_2O_2 -KF offers clear advantages over existing prior arts.
2. The nature of good stability of KF- H_2O_2 structure has also been reported (<https://doi.org/10.1023/A:1013040717382>), which is understandable given the strong hydrogen bond between F and H-O. However, the author introduced a concept of "borrowed blurring effect" but did not clarify the origin and meaning, or explain the chemical similarity in their finding.
3. To experimentally validated the stabilizing effect of KF- H_2O_2 , the author conducted Raman studies and claimed that the formation of strong hydrogen bond led to a shift in O-O peak from 874cm^{-1} (commercial) to 858cm^{-1} (KF- H_2O_2). However, in fig3, the signal-to-noise ratio for the KF- H_2O_2 sample in Fig. 3a is too low to distinguish the peak from background, which undermines the reliability of the interpretation of the Raman spectra.

Reviewer #3

(Remarks to the Author)

This manuscript presents an experimental and computational study of the direct solid-state H_2O_2 electrosynthesis and storage in the form of peroxosolvates of KF- H_2O_2 , $CO(NH_2)_2 \cdot H_2O_2$ and $Na_2CO_3 \cdot 1.5H_2O_2$. The as-fabricated solid-state H_2O_2 features not only high gravimetric densities but also good stability for repeated loading/deloading and shelf life. DFT calculations indicate the charge distribution of H-O and O-O bonds of H_2O_2 in KF- H_2O_2 becomes homogenizing compared with that of H_2O_2 molecule. Overall, this manuscript is interesting, and the results are properly presented and discussed. However, parts of the discussion lack evidences and more calculations are needed to support their conclusions. I therefore recommend a major revision before considering publication.

1. I think the stable load/deloading of H_2O_2 with KF, etc. is related the regulation of the H-O and O-O bond strength in H_2O_2 through the formation of possible hydrogen bond between H_2O_2 and KF. The "blurring effect" is not strongly related to the manuscript and should be revised to a more straightforward statement, such as "the regulation of bond length by the formation of possible hydrogen bond between H_2O_2 and F, N and O". I suggest the author revise the corresponding part of the manuscript.

2. I think the statement of "this band is shifted by 15cm^{-1} when compared to that of commercial H_2O_2 (873cm^{-1}), which is due to the operando hydrogen bonds between the electrochemically produced H_2O_2 and KF in the electrolyte" is too strong and should be revised. The formation of operando hydrogen bonds between H_2O_2 and KF in the electrolyte lacks direct evidence, and is just speculated from the operando Raman, according to the vibration frequency of O-O bond. H position generally is very difficult to be determined by XRD. Raman spectroscopy measures the vibrational mode but is also insufficient to detect the H position. Instead, neutron diffraction could determine the H position and related H bond, see *Inorganics* 2022, 10, 132. Here, I only suggest the author calculate the bond strength of the H bond (O-H...F, O-H...O and O-H...N), and the O-H bonds of KF- H_2O_2 , $CO(NH_2)_2 \cdot H_2O_2$ and $Na_2CO_3 \cdot 1.5H_2O_2$, and compare them with the O-H bond strength of isolated H_2O_2 molecule (a very related and nice paper could be referred and cited, *Inorganics* 2022, 10, 132), to provide an indirect supporting of this speculation.

3. Given that theory and multi-scale modeling and simulation, as supplements to experimental efforts, can help greatly to close some of the current experimental and technological gaps, as well as predict path-independent properties and help to fundamentally understand path-independent performance in multiple spatial and temporal scales, some relevant references are necessary to review to enhance the readability of the manuscript: e.g., *Chinese Physics B* 25, 018212 (2016).

4. The calculation of the bond strengths of H-O and O-O of H_2O_2 molecules in KF- H_2O_2 , $CO(NH_2)_2 \cdot H_2O_2$, $Na_2CO_3 \cdot 1.5H_2O_2$ and isolated H_2O_2 molecular is indispensable, because it could give a direct and quantitative data (instead of the very weak connection between the bond strength and charge transfer) to support their conclusions "inhibiting the break of these bondings inside H_2O_2 molecules".

Version 1:

Reviewer comments:

Reviewer #1

(Remarks to the Author)

The authors have thoroughly addressed all of my previous comments in a clear and satisfactory manner. Their point-by-point response demonstrates careful consideration and significant effort to improve the manuscript. I have no further concerns, and I recommend the manuscript for acceptance in its current form.

Reviewer #2

(Remarks to the Author)

The reviewer appreciated the author's effort in providing a clearer discussion on the process and mechanism. However, several concerns remained in the presentation of performance and mechanism.

1. The proposed and described novel H₂O₂ stabilization mechanism: "blurring effect" is conceptually the same as forming a stronger hydrogen bond network that is widely applied in the field of stabilizing H₂O₂. It is recommended that the author present the conceptual advancement beyond the previous knowledge.

2. The yield of H₂O₂ and multiple related performance parameters were evaluated by quantifying the H₂O₂ concentration in the electrolyte. While the major technical advancement was given to the formation of solid-state KF-H₂O₂, the H₂O₂ yield should also be analyzed in the final solid form to provide a complete analysis in case of potential H₂O₂ loss during the solidification.

3. Several important testing parameters were missing from the discussion and the method section. For example, the H₂O₂ concentration in the electrolyte before evaporation and solidification, and the air flow rate used at each current density for the flow cell tests. The fabrication method of the catalyst-coated carbon fiber paper and how the catalyst loading was determined. These parameters were essential in evaluating normalized performance in electrocatalysis.

4. An impressive production rate was presented in Figure 4d (>250 mol/g/h). However, the reviewer had difficulties in the calculation process since the scaling up of device area should not benefit the mass-specific production rate (total catalyst loading should also be scaled up). Therefore, using 3.5A, 0.2mg/cm², 89.6%FE, 16cm² should give a production rate of 18.28 instead of 292.35 mol/g/h, which is 16x of this value.

Reviewer #3

(Remarks to the Author)

Authors have revised the manuscript point by point. I recommend that the present version be accepted for publication.

Version 2:

Reviewer comments:

Reviewer #2

(Remarks to the Author)

The authors have addressed my concerns and corrected the reported performance. I recommend the manuscript for publication in its current form

We would like to thank the Editor's invitation to revise the manuscript. We are also appreciated of the reviewer #1 for his/her positive recommendation and constructive comments to improve the quality of this work.

After receiving the reviewer's comments, we have spent several intensive weeks in order to answer the concerns raised by the astute reviewer. We believe we have shifted the work to the next level of proof and gained a much deeper understanding of the system. Attached please see the point-by-point response to the reviewer #1's comments. Thank you and best wishes,

Kind regards

The authors

List of changes

- (1) We have replaced the images with high-resolution ones. (**Response to Reviewer #1 Comment 1**; Figs. 1–4)
- (2) We have modified/added descriptions to corresponding images and data. (**Response to Reviewer #1 Comment 2**; Page 5 line 13–23, 17–20, 21–23; Page 6 line 11–12; Page 6 line 22–25; Page 11 line 23–24; Supplementary Figures 3, 9, 21–23, 46 and Supplementary Table 1)
- (3) We have corrected the description of relevant contents. (**Response to Reviewer #1 Comment 3**; Page 4 line 21–22; Page 5 line 13–17)
- (4) We have supplemented with bond energy data for H₂O₂ and peroxosolvates (**Response to Reviewer #1 Comment 4**; Supplementary Figures 33–34; Page 8 line 24; Page 9 line 1–10; Reference 27)
- (5) We have explained the hydrogen-bonding interactions between H and F/N/O using FTIR data. (**Response to Reviewer #1 Comment 5**; Supplementary Figure 3)
- (6) We have corrected typos and errors in corresponding images. (**Response to Reviewer #1 Comment 6**; Figure 2)
- (7) We have performed additional tests on *operando* Raman spectra. (**Response to Reviewer #1 Comment 7**; Page 8 line 14–15; Supplementary Figure 32; Supplementary Note 5; Supplementary information reference 11–13)
- (8) We have explained the blurring effect in details and add a 2e-ORR test for NaF and K₂CO₃ electrolytes. (**Response to Reviewer #1 Comment 8**; Page 9 line 22–24; Page 10 line 1–7; Supplementary Figure 38)
- (9) We have rechecked and discussed L-H and E-R models. (**Response to Reviewer #1 Comment 9**)

Response to Reviewer #1

Original Comment: *This paper presents a solid-state electrosynthesis approach for H_2O_2 production, demonstrating a productivity that exceeds that of the conventional industrial process. Traditional H_2O_2 synthesis requires strategies to suppress its decomposition; in this study, the authors introduce peroxosolvates to mitigate H_2O_2 degradation, achieving a productivity of $0.943 \text{ mol L}^{-1} \text{ h}^{-1}$, which surpasses that of the benchmark anthraquinone method. The work is significant as it proposes a novel electrochemical system design for efficient peroxide production. I recommend this manuscript for publication with minor revisions. Specific comments are as follows.*

Original Comment 1. *The overall image quality is not satisfactory. It is necessary to enhance the resolution of the figures to improve readability.*

Response: Thanks for your kind reminding. We have replaced the original images in the manuscript with high-resolution ones as follows:

Figure 1 | Schematic illustration of H_2O_2 production. **a**, Direct solid-state H_2O_2 electrosynthesis in flow cells. **b**, Overview of the development for H_2O_2 economy. **c**, Sketch of solid-state H_2O_2 electrosynthesis in the form of peroxosolvates at

three-phase boundary. **d**, Traditional and the present production diagrams for solid-state H_2O_2 . **e**, XRD patterns of $\text{KF}\cdot\text{H}_2\text{O}_2$ as comparison to KF . **f**, Theoretical H_2O_2 storage percentages for peroxosolvates like $\text{KF}\cdot\text{H}_2\text{O}_2$, $\text{CO}(\text{NH}_2)_2\cdot\text{H}_2\text{O}_2$ and $\text{Na}_2\text{CO}_3\cdot 1.5\text{H}_2\text{O}_2$. **g–h**, Changes in accumulative H_2O_2 storage capacity and cost for $\text{KF}\cdot\text{H}_2\text{O}_2$ after 100 cycles or 160 days.

Figure 2 | Direct solid-state H_2O_2 electroynthesis. **a**, High-resolution transmission electron microscopy (HR-TEM) image of ZIF-350 catalyst (scale bar: 200 nm). **b**, X-ray diffraction (XRD) pattern of ZIF-350 catalyst. **c**, RRDE linear sweep voltammetry (LSV) of ZIF-350 catalyst recorded at a rotation rate of 1600 rpm, along with the detected H_2O_2 currents on a Pt ring electrode at a fixed potential of 1.2 V vs. RHE. **d–e**, Faradaic efficiency (FE) and yield rates in flow cells. **f**, The current densities at low O_2 -content environments in flow cells. **g**, Cumulative H_2O_2 concentrations in flow cells. **h**, The stability test at 200 mA cm⁻² in flow cells, with the inset showing the XRD patterns of catalyst before and after test.

Figure 3 | Mechanistic investigation. **a**, The operando Raman spectra for air-to-H₂O₂ conversion. **b**, Adsorption energy of KF with H₂O₂ and H₂O (inset shows the charge transport inside KF*H₂O₂). **c**, H₂O₂ yield rates in different KF concentrations in flow cells. **d**, Free-energy diagrams for 2 and 4e-ORR pathways. **e**, The theoretical volcano curve for 2e-ORR. **f**, Kinetic barriers for *O₂ → *OOH via Eley-Rideal (E-R) and Langmuir-Hinshelwood (L-H) mechanisms. **g**, Schematic air-to-H₂O₂ solidification pathway.

Figure 4 | Scale-up production. **a**, Schematic configuration of our developed tandem/parallel prototype device. **b**, The optical picture of the prototype device. **c**, The H_2O_2 yield rates under ampere-level currents. **d**, The comparison of yield rates with the state-of-the-art literature for air-to- H_2O_2 conversion (please also see Supplementary Table 9). **e**, XRD patterns of electro-synthesized $\text{KF}\cdot\text{H}_2\text{O}_2$ as comparison to commercial counterpart. **f**, The stability test at 1 A for 50 hrs. The red arrows represent electrolyte refresh. **g**, Potential application of $\text{KF}\cdot\text{H}_2\text{O}_2$ for the degradation of methylene blue pollutants.

Original Comment 2. *In many cases, the figure descriptions are either missing, insufficient, or inaccurate.*

For example:

- There is no description provided for Figures 1a and 1b.

Response: We have added the descriptions as follows:

Experimentally, starting in an air-to-H₂O₂ conversion system in a flow-type cell (Fig. 1a), we have explored each possible parameter of local environments (like pHs²³, solvents²⁴, wettability²⁵ and electric fields²⁶), and unexpectedly observed *operando* hydrogen bonds formed between solutes (*i.e.*, KF) and local H₂O₂ molecules, which enables *in situ* electrosynthesis of solid-state H₂O₂. Notably, O₂ from air diffuses through gas diffusion layer, adsorbs and is activated on the catalyst surface, which then couples with protons (H⁺) generated from H₂O electrolysis to form H₂O₂. The *operando* formation of solid-state H₂O₂ has been achieved generally through its reaction with KF, urea, and Na₂CO₃ *via* hydrogen bonding, leading to the formation of peroxosolvates (Fig. 1c). By circumventing the instability of H₂O₂, this method offers a strategy that eliminates the need for the cumbersome and energy-intensive steps of traditional indirect solid H₂O₂ synthesis, enabling direct *operando* electrosynthesis of solid-state H₂O₂ (Figs. 1b, d). (Page 5 line 13–23)

- Additional and more specific explanations are needed for the supplementary data.

Response: We have added more specific explanations on supplementary data as follows:

Supplementary Figure 3. FTIR spectra of peroxosolvates. a, KF and KF·H₂O₂. **b,** CO(NH₂)₂ and CO(NH₂)₂·H₂O₂. **c,** Na₂CO₃ and Na₂CO₃·1.5H₂O₂. Generally, KF alone only show typically K-F vibration at 603 cm⁻¹ and O-H at 3292 cm⁻¹. While for KF·H₂O₂, both vibrations have changed significantly, *i.e.*, 571 cm⁻¹ for K-F and 3036 cm⁻¹ for O-H vibration. The peak shift might be originated to the formation of O-H··F bonds. The same phenomenon has been observed for CO(NH₂)₂·H₂O₂ and Na₂CO₃·1.5H₂O₂ counterparts.

Supplementary Figure 9. Additional TEM images of ZIF-350. The TEM images illustrate that ZIF-350 possesses a rhombic dodecahedral structure with particle sizes ranging from 100 to 200 nm.

Supplementary Figure 21. LSV curves of ZIF-350 tested in a flow cell system in air. ZIF-350 electrode has exhibited a well-defined LSV profile in KF electrolyte, where the current density rapidly increases with negatively polarized potentials, suggesting the occurrence of oxygen reduction reaction (ORR) in air atmosphere.

Supplementary Figure 22. Standard calibration curves for quantifying H₂O₂. **a**, UV-vis spectra of I³⁻ solution with various concentrations. **b**, The corresponding standard curve. The H₂O₂ concentration was quantified by iodometric method. The UV-vis absorbance of the resulted mixture was measured at the wavelength of 351 nm.

Supplementary Figure 23. UV-vis spectra curves of ZIF-350 tested in different current densities in flow cell system in air. The air-to-H₂O₂ performance of ZIF-350 in KF was examined by UV-vis spectra, showing the concentration of H₂O₂ increases almost linearly throughout whole current density window.

- In Figure 1g, it is unclear why the storage capacity improves with increasing cycle number; this requires further clarification.

Response: We are sorry for not explaining it clearly. The storage capacity has been mentioned in Figure 1g, which refers to the accumulative H₂O₂ storage capacity of KF. More specifically, in a typical cycle for KF, the low-temperature heating of KF·H₂O₂ simply releases H₂O₂, and then KF is left to be used again for further H₂O₂ storage. As the cycle number increases, more and more H₂O₂ can be stored by KF, so the accumulative H₂O₂ storage capacity increased with cycle number.

- The purpose of Figure 1f is not clearly explained.

Response: Figure 1f illustrates the H₂O₂ storage mass fractions of KF, CO(NH₂)₂ and Na₂CO₃ as follows:

Mass ratio calculations show the nominal H₂O₂ gravimetric densities of 36.9 wt% for KF·H₂O₂, 36.2 wt% for CO(NH₂)₂·H₂O₂ and 32.5 wt% for Na₂CO₃·1.5H₂O₂, respectively (Fig. 1f). (Page 6 line 11–12)

- For Figures 1c and 1d, the schemes are not clearly described, making it difficult to understand what they represent.

Response: Thanks for the kind reminding. We have added relevant description as follows:

Notably, O₂ from air diffuses through gas diffusion layer, adsorbs and is activated on the catalyst surface, which then couples with protons (H⁺) generated from H₂O electrolysis to form H₂O₂. The *operando* formation of solid-state H₂O₂ has been achieved generally through its reaction with KF, urea, and Na₂CO₃ *via* hydrogen bonding, leading to the formation of peroxosolvates (Fig. 1c). (Page 5 line 17–20)

By circumventing the instability of H₂O₂, this method offers a strategy that eliminates the need for the cumbersome and energy-intensive steps of traditional indirect solid H₂O₂ synthesis, enabling direct *operando* electrosynthesis of solid-state H₂O₂ (Figs. 1b, d). (Page 5 line 21–23)

- The explanation for Figure/Table 1 is insufficient.

Response: Supplementary Table 1 presents the mass change of KF·H₂O₂ with cycle numbers. After 100 cycles, KF still retains its ability to store H₂O₂. We have added a more description to Supplementary Table 1 as follows:

Supplementary Table 1. Number of cycles for KF to store and release H₂O₂. In a typical cycle for KF, the low-temperature heating of KF·H₂O₂ releases H₂O₂, where KF is left for further storage.

Cycle number	Mass of KF·H ₂ O ₂ (g)
1	1.59
10	1.57
20	1.58
30	1.58
40	1.57
50	1.57
60	1.58
70	1.58
80	1.59
90	1.59

- While the storage capacity appears to improve with an increasing number of cycles, this trend is not addressed in the text and should be discussed.

Response: We are sorry for not explaining it clearly. The storage capacity has been mentioned in Figure 1g, which refers to the accumulative H₂O₂ storage capacity of KF. More specifically, in a typical cycle for KF, the low-temperature heating of KF·H₂O₂ simply releases H₂O₂, and then KF is left to be used again for further H₂O₂ storage. As the cycle number increases, more and more H₂O₂ can be stored by KF, so the accumulative H₂O₂ storage capacity increased with cycle number. We have added the follow descriptions in the main text:

The H₂O₂ storage capacity of KF increases linearly from 0.587 H₂O₂ kg⁻¹/KF kg⁻¹ to 58.73 H₂O₂ kg⁻¹/KF kg⁻¹. Correspondingly, the cost per unit mass of KF decreases sharply from 1.2 to 0.06 \$ KF/\$ H₂O₂ (after 20 cycles) and then gradually to 0.012 \$ KF/\$ H₂O₂ (after 100 cycles). (Page 6 line 22–25)

- There is no explanation provided for Figures 21 through 23.

Response: We have added the relevant descriptions as follows:

Supplementary Figure 21. LSV curves of ZIF-350 tested in a flow cell system in air. ZIF-350 electrode has exhibited a well-defined LSV profile in KF electrolyte, where the current density rapidly increases with negatively polarized potentials,

suggesting the occurrence of oxygen reduction reaction (ORR) in air atmosphere.

Supplementary Figure 22. Standard calibration curves for quantifying H_2O_2 . **a**, UV-vis spectra of I_3^- solution with various concentrations. **b**, The corresponding standard curve. The H_2O_2 concentration was quantified by iodometric method. The UV-vis absorbance of the resulted mixture was measured at the UV-vis wavelength of 351 nm.

Supplementary Figure 23. UV-vis spectra curves of ZIF-350 tested in different current densities in flow cell system in air. The air-to- H_2O_2 performance of ZIF-350 in KF was examined by UV-vis spectra, showing the concentration of H_2O_2 increases almost linearly throughout whole current density window.

- In Figure 40, the content of the illustration seems inconsistent with its description.

Response: We are sorry for making the mistakes. We have corrected the error as follows:

Under these conditions, the 2e-ORR activity has been quantified through a chronoamperometric tests at different applied currents (Fig. 4c, Supplementary Fig. 46). (Page 11 line 23–24)

Supplementary Figure 46. UV-vis spectra curves of ZIF-350 tested in the present system under air. ZIF-350 maintains exceptional 2e-ORR performance even at ampere-level currents. UV-vis spectra confirm the H_2O_2 concentration increasing linearly with current density.

Overall, many parts of the figures and their corresponding explanations require revision and clarification.

Response: We are appreciated of the reviewer for his/her kind help. Beside above errors and typos, we have further rechecked the whole manuscript.

Original Comment 3. *Additional proofreading is necessary to correct typographical errors and verify the accuracy of the content. For example, in line 71, it should be confirmed whether " H_2O " is the correct term in the phrase "direct air-to-diluted H_2O (<10 wt%)". Also, in line 86, it is unclear whether the reference to "Fig. 1c and d" is accurate. A more thorough review of such details is recommended.*

Response: Thanks for the kind reminding. We have corrected the error and typos as follows:

Recently, we and other groups have addressed this problem by reporting direct air-to-diluted H_2O_2 (< 10 wt%) conversion at high selectivity by mediating gas diffusion electrodes. (Page 4 line 21–22)

Experimentally, starting in an air-to- H_2O_2 conversion system in a flow-type cell (Fig. 1a), we have explored each possible parameter of local environments (like pHs²³, solvents²⁴, wettability²⁵ and electric fields²⁶), and unexpectedly observed *operando* hydrogen bonds formed between solutes (*i.e.*, KF) and local H_2O_2 molecules, which enables *in situ* electrosynthesis of solid-state H_2O_2 . (Page 5 line 13–17)

Original Comment 4. *The term "blurring effect" needs to be explained in more detail and with greater clarity regarding its meaning and implications. If possible, including a simulation that calculates and compares the bond dissociation energies of H–H and O–H would further strengthen the explanation.*

Response: We are appreciated of the reviewer for his/her constructive comment. Generally, the blurring effect describes hydrogen-bond-induced charge delocalization. Analogous to Gaussian blurring in image processing averages pixel intensities to reduce local contrast, here we explain this effect as homogenizing electron distribution across H-O and O-O bonds within H_2O_2 molecules *via* *operando* hydrogen-bond networks (*e.g.*, $\text{KF}\cdot\text{H}_2\text{O}_2$ formation), thereby suppressing H_2O_2 decomposition.

Following the reviewer's suggestion, we have calculated the bond energies for isolated H_2O_2 and H_2O_2 in peroxosolvates to further explain blurring effect as follows:

To further validate the stabilizing role of blurring effect on H_2O_2 molecule, we have calculated hydrogen-bond energies for isolated H_2O_2 and H_2O_2 in peroxosolvates (*i.e.*, $\text{KF}\cdot\text{H}_2\text{O}_2$, $\text{CO}(\text{NH}_2)_2\cdot\text{H}_2\text{O}_2$ and $\text{Na}_2\text{CO}_3\cdot 1.5\text{H}_2\text{O}_2$, Supplementary Fig. 33)²⁷. Generally, the instability of H_2O_2 molecule originates from nonuniform charge distributions between different bonds, with a bonding energy difference of 1.07 eV between O-H and O-O bonds. Interestingly, both bond energies of O-H and O-O in peroxosolvate have been reduced, attributable to bond elongation induced by

hydrogen bonding (i.e., O-H \cdots N, O-H \cdots O and O-H \cdots F, Supplementary Fig. 34). The difference between the O-H and O-O bond energies in peroxosolvates has also been reduced to 0.90 eV for KF \cdot H₂O₂, 0.91 eV for CO(NH₂)₂ \cdot H₂O₂ and 0.92 eV for Na₂CO₃ \cdot 1.5H₂O₂, respectively (Supplementary Fig. 33d). Accordingly, we consider the hydrogen bonding has introduced blurring effect between O-H and O-O bonds, enabling high stability of H₂O₂ molecule. (Page 8 line 24, Page 9 line 1–10)

Supplementary Figure 33. Bond energies of isolated H₂O₂ and H₂O₂ in peroxosolvates. a, Bond energies of O-H \cdots N/O/F. b, Bond energies of O-H. c, Bond energies of O-O. d, Bond energy difference between O-H and O-O bonds.

Supplementary Figure 34. Bond length of isolated H₂O₂ and H₂O₂ in peroxosolvates. a, Bond length of O-H in H₂O₂. b, Bond length of O-O in H₂O₂. c, Schematic of stabilization of H₂O₂ by blurring effect.

Original Comment 5. *In line 104, it is unclear why H is considered to be bonded with FNO. When explaining this using FTIR data, it would be helpful to include the specific peak assignment that supports this interpretation.*

Response: Thanks for the reviewer for his/her constructive comment. According to its definition (Chem. Phys. Lett. 463, 1–3 (2008)), hydrogen bonding is a special type of intermolecular force, formed generally by electrostatic attraction between a hydrogen atom and those atoms with large electronegativity (typically F, N, O). Therefore, H can bond with F/N/O easily in peroxosolvates theoretically.

Experimentally, we have added FTIR test to explain the hydrogen bonding between a hydrogen atom and F/N/O. As shown in Supplementary Figure 3a, KF alone only show typically K-F vibration at 603 cm⁻¹ and O-H at 3292 cm⁻¹. While in KF·H₂O₂, both vibrations have changed significantly, *i.e.*, 571 cm⁻¹ for K-F and 3036 cm⁻¹ for O-H vibration. According to the response to Comment 4, the peak shift is originated to the formation of O-H···F bonds. The same phenomenon has been observed for CO(NH₂)₂·H₂O₂ and Na₂CO₃·1.5H₂O₂ counterpart.

Supplementary Figure 3. FTIR spectra of peroxosolvates. a, KF and KF·H₂O₂. **b,** CO(NH₂)₂ and CO(NH₂)₂·H₂O₂. **c,** Na₂CO₃ and Na₂CO₃·1.5H₂O₂. Generally, KF alone only show typically K-F vibration at 603 cm⁻¹ and O-H at 3292 cm⁻¹. While for KF·H₂O₂, both vibrations have changed significantly, *i.e.*, 571 cm⁻¹ for K-F and 3036 cm⁻¹ for O-H vibration. The peak shift might be originated to the formation of O-H··F bonds. The same phenomenon has been observed for CO(NH₂)₂·H₂O₂ and Na₂CO₃·1.5H₂O₂ counterparts.

Original Comment 6. *In Figure 2f, the experimental data for the 10% condition shows a higher current density; however, the corresponding yield appears to be lower. This discrepancy should be addressed or clarified.*

Response: Thanks for your kind reminding. We have re-tested the experiments for 10% O₂ condition, and re-calculated the yield rate as follows:

Generally, the H₂O₂ yield rate (mol g_{cat}⁻¹ h⁻¹) is calculated at 160 mA cm⁻² according to the following equation:

$$Yield = \frac{C_{H_2O_2} V_{electrolyte}}{t m_{cat}} = \frac{0.00507 \text{ mol L}^{-1} \times 40 \text{ mL}}{\frac{1}{12} \text{ h} \times 0.2 \text{ mg cm}^{-2}} = 12.16 \text{ mol g}_{cat}^{-1} \text{ h}^{-1}$$

Where $C_{H_2O_2}$ is H₂O₂ concentration (mol L⁻¹), $V_{electrolyte}$ is the volume of cathodic electrolyte (L), m_{cat} is catalyst mass loading (mg cm⁻²)

The limiting current density of 2e-ORR at 10% O₂ concentration is 163 mA cm⁻² as deduced from the formula (7) in Method section, and at an experimental current density of 160 mA cm⁻², the H₂O₂ yield is only 12.16 mol g_{cat}⁻¹ h⁻¹, which is below the limiting value (Fig 2f). We have updated the data in corresponding figures.

Figure 2 | Direct solid-state H₂O₂ electrosynthesis. **a**, High-resolution transmission electron microscopy (HR-TEM) image of ZIF-350 catalyst (scale bar: 200 nm). **b**, X-ray diffraction (XRD) pattern of ZIF-350 catalyst. **c**, RRDE linear sweep voltammetry (LSV) of ZIF-350 catalyst recorded at a rotation rate of 1600 rpm, along with the detected H₂O₂ currents on a Pt ring electrode at a fixed potential of 1.2 V vs. RHE. **d–e**, Faradaic efficiency (FE) and yield rates in flow cells. **f**, The current densities at low O₂-content environments in flow cells. **g**, Cumulative H₂O₂ concentrations in flow cells. **h**, The stability test at 200 mA cm⁻² in flow cells, with the inset showing the XRD patterns of catalyst before and after test.

Original Comment 7. *In line 149, the Raman peak at 0 hours is not clearly visible. Similarly, the peak at 4 hours mentioned in line 151 also appears to be indistinct and*

should be re-examined.

Response: Thanks for your kind reminding. We have re-tested the *operando* Raman data for ten times and provide the responses as follows:

1) At the beginning of the test (0 hour), there was no H₂O₂ generated but only KF in the electrolyte, so only a weak Raman signal could be observed due to the small Raman scattering cross section of the ionic KF compound.

2) For *Raman peak at 4 hours*, we consider the weak KF·H₂O₂ signal due to harsh *operando* test conditions during experimental condition. By further reviewing the relevant literature (Phys. Chem. Lett. 2022, 13, 479), we found the following reasons:

Generally, Raman spectra is a scattering signal with intrinsic weak intensity. When used to study the structural characteristics of a bulk material, obvious signal peaks can be seen. This is due to the structural crystal lattice that contributes to overall vibration of the bulk material, leading to enhanced scattered signals.

While in this work, the *operando* Raman only probes the signals of O-O stretching vibrations in H₂O₂ during catalytic processes. The scattered Raman signals are mainly focused on bond vibrations of adsorbed species (like *OOH in 2e-ORR) on catalyst surfaces, which are known to be very weak as comparison to structural crystal lattices in bulk materials. Consequently, the *operando* Raman signals for catalytic reactions is mostly very weak in the literatures (like ORR, NRR, CRR). *J. Am. Chem. Soc.* 142, 715-719 (2020); *Angew. Chem. Int. Edit.* 60, 20331-20341 (2021); *ACS Nano* 14, 11363-11372 (2020) Our *operando* Raman signals in Figure 3a and Supplementary Figure 32 are comparable to the above literature.

To confirm the accuracy of the experimental results, we performed 10 *operando* Raman tests. We directly determined the positions and intensities of the peaks using Raman spectrometer software, and updated the *operando* Raman data with error bars. By comparing the the Raman vibration of ZIF-350 at different electrolysis times, we conclude the intensity of *OOH peak increased consistently with elongated reaction duration, proving the validity of our *operando* synthesis condition.

The specific changes are as follows:

After 2-hr electrolysis, a Raman vibration has emerged at 858 cm^{-1} , whose peak intensity increases with elongated electrolysis at 4 hrs (Fig. 3a and Supplementary Fig. 32). (Fig. 3a and Supplementary Fig. 32) (Page 8 line 14–15)

Supplementary Figure 32. Ten repetitive *operando* Raman tests for ZIF-350 in air under the same condition. a, 4 hrs. b, 2 hrs. c, 0 hr. d, The 858 cm^{-1} peak intensity with error bar in *operando* Raman test. Ten measurements were conducted for each data point with the error bars corresponding to the standard deviation.

Supplementary Note 5:

Additional ten repetitive tests were performed for *operando* Raman under the same condition of Supplementary Figure 32. The weak Raman signal is due to the harsh *operando* test condition¹¹. In this work, the *operando* Raman only probes the signals of O-O stretching vibrations in H_2O_2 during catalytic processes. The scattered Raman signals are mainly focused on bond vibrations of adsorbed species (like $^*\text{O}_2$ and $^*\text{OOH}$ in ORR) in electrolyte, which are known to be very weak as comparison to bulk materials. To confirm the accuracy of the experimental results, Raman

spectrometer software is used to directly determine the peak positions and intensities. On this basis, the Raman information of H₂O₂ in KF are compared at different durations. By comparing the Raman vibration of ZIF-350 at different electrolysis durations^{12, 13}, we conclude the intensity of *OOH peak increased consistently with elongated reaction duration, proving the validity of our operando synthesis condition.

Original Comment 8. *The common characteristics of the additives used in this study—such as KF, Na₂CO₃, and urea—are not clearly explained. It would be helpful to clarify whether the blurring effect would still occur if alternative salts such as NaF were used instead of KF, or Na₂CO₃ instead of Na₂CO₃. Furthermore, the specific conditions under which the blurring effect occurs should be described in more detail, ideally in relation to the physicochemical properties of the additives.*

Response: We are appreciated of the constructive comment. Our response is as follows:

1) Firstly, the common property of KF, Na₂CO₃ and urea additives is the formation of hydrogen bonds with H₂O₂ molecules. These hydrogen bonds can inhibit H₂O₂ decomposition by homogenizing the charge distribution of the H-O and O-O bonds through blurring effect. More specifically, F in KF, CO₃²⁻ in Na₂CO₃ and NH₂ in urea can form hydrogen bonds with H atom in H₂O₂. We found that the adsorption energy of KF*H₂O₂ is -0.90 eV, which is significantly lower than that of KF*H₂O (-0.50 eV) by DFT calculations, suggesting that the hydrogen bonding between KF and H₂O₂ is strong. Such strong interaction can blur the charge distribution of the O-H and O-O bonds thus the stabilization of H₂O₂ molecules.

2) To verify the universality of the blurring effect, we have conducted experiment tests in NaF and K₂CO₃ electrolytes as follows:

The blurring effect is not exclusive to KF. Our additional tests using NaF and K₂CO₃ as electrolytes were conducted in the same condition. For NaF electrolyte, the H₂O₂ Faradaic efficiencies (FEs) of ZIF-350 consistently exceeded 90% across the entire range of applied current densities, the values being 97.5%~90.8% at 50~350 mA cm⁻², respectively (Supplementary Fig. 38a). Even at the theoretical limiting

current density of air-to-H₂O₂ solidification ($\sim 350 \text{ mA cm}^{-2}$), the FE reaches 90.8%, together with the exceptional H₂O₂ yield rate of $29.62 \text{ mol g}_{\text{cat}}^{-1} \text{ h}^{-1}$. On the other hand, for K₂CO₃ electrolyte, the H₂O₂ Faradaic efficiencies (FEs) of ZIF-350 is 97.9%~87.8% at 50~350 mA cm⁻² (Supplementary Fig. 38b), and reaching 87.8% and $28.64 \text{ mol g}_{\text{cat}}^{-1} \text{ h}^{-1}$ at $\sim 350 \text{ mA cm}^{-2}$. These results suggest that electrolytes capable of forming hydrogen bonds with H₂O₂ can also introduce blurring effect. (Page 9 line 22–24, Page 10 line 1–7)

Supplementary Figure 38. The 2e-ORR performances of ZIF-350 in NaF and K₂CO₃ electrolytes. a, 0.5 M NaF. b, 1.0 M K₂CO₃.

3) As mentioned above, the blurring effect requires additives capable of forming hydrogen bonds (e.g., O-H···F, O-H···O, O-H···N) with H₂O₂, where the electronegative atom (F/O/N) homogenizes charge distribution across O-O and O-H bonds. As a consequence, whenever H₂O₂ is generated at the local environment of electrode-electrolyte interface, it interacts with the additive, which is crucial for the stabilization.

Original Comment 9. *In Figure 37, it is unclear whether the illustrations of the Er and lh models are accurate. This should be carefully reviewed for correctness.*

Response: We are appreciated of the reviewer for his/her constructive suggestion. We give the responses as follows:

1) According to the literature (Nat. Chem. 11, 722-729 (2019)), the classical L-H mechanism can be summarized as follows: both reactant gas molecules A and B need to be adsorbed on the surface of the catalyst (*A and *B), and the adjacent *A and *B

couple to form the next product $*AB$. In contrast, the E-R mechanism only requires the adsorption of a single reactant gas molecule (e.g., $*A$), and $*A$ on the surface combines with the free gas molecule B in the surrounding environment to form $*AB$. JACS Au 3, 943-952 (2023)

2) In this work, we have updated the L-H and E-R mechanisms in three-phase boundary condition for 2e-ORR. The L-H mechanism is updated as gaseous A and the free substance B are adsorbed on the catalyst surface to form $*A-*B$, respectively, and then the free substance B is in the process of combining with $*A$ to finally form the product of $*AB$; the E-R mechanism is updated as gaseous A is firstly adsorbed on the catalyst surface to form $*A$, followed by the free species B in the liquid phase directly combining with $*A$ at triple-phase interface, finally forming the product of $*AB$.

3) Accordingly, we have built the catalyst model for updated L-H and E-R mechanisms, starting by the L-H mechanism depicted as the adsorption of $*O_2$ onto the Zn active site and the adsorption of $*H$ onto an adjacent oxygen site. In contrast, E-R mechanism is represented by the direct adsorption of $*OOH$ onto the Zn active site.

Upon calculating the Gibbs free energy, we discover that $*O_2$ and $*H$ have poor adsorption on the surface of ZIF-350. And the L-H mechanism pathway ($*O_2 \rightarrow *O_2-*H \rightarrow *OOH$) has a positive free energy change of 0.59 eV, indicating a high reaction energy barrier (Supplementary Fig. 43), causing the L-H mechanism difficult to proceed. Comparably, the E-R mechanism pathway ($*O_2 \rightarrow *OOH$) has a negative free energy change of -0.69 eV and tends to react spontaneously. Therefore, ZIF-350 shows a preference for the Eley-Rideal mechanism to Langmuir-Hinshelwood counterpart.

We would like to thank the Editor's invitation to revise the manuscript. We are also appreciated of the reviewer #2 for his/her constructive comments to improve the quality of this work.

After receiving the reviewer's comments, we have spent several intensive weeks in order to answer the concerns raised by the astute reviewer. We believe we have shifted the work to the next level of proof and gained a much deeper understanding of the system. Attached please see the point-by-point response to the reviewer #2's comments. Thank you and best wishes,

Kind regards

The authors

List of changes

- (1) We have explained the advantages and disadvantages of KF as a carrier for solid-state storage of H₂O₂ and extend the system to urea and Na₂CO₃ counterparts. (**Response to Reviewer #2 Comment 1**; Page 8 line 7–9; Supplementary Figure 31)
- (2) We have discussed the concept of blurring effect. (**Response to Reviewer #2 Comment 2**; Page 8 line 24; Page 9 line 1–10; Page 9 line 22–24; Page 10 line 1–7; Supplementary Figures 33–34; Supplementary Figure 38 and Reference 27)
- (3) We have conducted repetitive tests on the *operando* Raman Spectra. (**Response to Reviewer #2 Comment 3**; Supplementary Figure 32; Supplementary Note 5; Supplementary information reference 11–13)

Response to Reviewer #2

Original Comment: *The author describes the use of the strong hydrogen-bonding interaction between KF and H₂O₂ to stabilize the H₂O₂ in a solid state KF-H₂O₂ with H₂O₂ concentration >30wt%. While achieving a high H₂O₂ concentration and the enhanced stability are valuable and important metrics for practical applications, the underlying concept of utilizing hydrogen bond to stabilize H₂O₂ to form solid-state crystalline peroxosolvates has already been extensively explored, in particular, KF has been identified to be one of the most stable form of peroxosolvate hosts. Accordingly, I have the following major concerns regarding the novelty, experimental validation of the proposed mechanism, and the practical applicability of the proposed process.*

Original Comment 1. *The concept of crystalline peroxosolvates has been widely adopted in the industrial process, where the Na₂CO₃-H₂O₂ and Urea-H₂O₂ have been the widely adopted options given their less toxic and less hazardous nature. By contrast, the use of KF in the process may raise environmental and safety concerns that could limit the practical applicability of this process. It is suggested that the author provides a more in-depth discussion of the sustainability considerations associated with KF. Or please identify a specific use case in which H₂O₂-KF offers clear advantages over existing prior arts.*

Response: We fully understand the environmental and safety concerns raised by the reviewer. Our response is as follows:

1) Firstly, it is true that fluorine in KF can be toxic and corrosive, which may cause potential environmental and safety impacts. While nowadays KF has been widely applied in various technical fields, such as the production of specialty glasses (Renewable Energy 203, 56–67 (2023)), ceramics (Mater. Chem. Phys. 192, 304–310 (2017)), and pharmaceuticals (Acc. Chem. Res. 2020, 53, 2, 322–334). Its stability and cost-effectiveness make it a popular choice in versatile industrial applications. There are already many well-developed personal protective equipment for handling and

processing KF, for example, safety glass, protective eyewear and gloves. The safety concerns related to KF can be effectively managed with proper safety procedures and precautions.

2) Next, in our work, KF has been utilized as a carrier for H_2O_2 , owing to its exceptional stability and strong hydrogen-bonding interactions. Our experiments demonstrate $\text{KF}\cdot\text{H}_2\text{O}_2$ can undergo over 100 loading-deload cycles or 160 days (Fig. 1g–h). This has led to a large storage capacity of $58.73 \text{ H}_2\text{O}_2 \text{ kg}^{-1}/\text{KF kg}^{-1}$ and low cost of $0.012 \text{ \$ KF}/\text{\$ H}_2\text{O}_2$ (after 100 cycles). Therefore, only minimal quantity of KF is required for H_2O_2 storage which could significantly reduce the potential environmental impact. The recyclability of KF ensures the overall environmental footprint of the process minimized, making it a possible option for H_2O_2 storage.

3) More importantly, in the main text and supporting information (Fig. 1 and Supplementary Figs. 1–6), we have further demonstrated the synthetic procedure is general, and has been readily extended to other peroxosolvates such as $\text{CO}(\text{NH}_2)_2\cdot\text{H}_2\text{O}_2$ and $\text{Na}_2\text{CO}_3\cdot 1.5\text{H}_2\text{O}_2$. Herein, by adopting similar methodologies, we have successfully synthesized $\text{Na}_2\text{CO}_3\cdot 1.5\text{H}_2\text{O}_2$ and $\text{CO}(\text{NH}_2)_2\cdot\text{H}_2\text{O}_2$ as well (Supplementary Fig. 6). The detailed data are presented as follows:

Supplementary Figure 6. Changes in H_2O_2 storage capacity and cost by urea and Na_2CO_3 for 10 storage-release cycles. Where x stands for urea and Na_2CO_3 .

Notably, the stabilizing mechanism observed with KF can be readily extended to

other systems such as urea and Na_2CO_3 , demonstrating the generality of this approach (Supplementary Fig. 31). (Page 8 line 7–9)

Supplementary Figure 31. The 2e-ORR performances in urea and Na_2CO_3 . a, 0.5 M NaCl. b, 0.5 M NaCl with 1.0 M urea. c, 0.5 M K_2SO_4 . d, 0.5 M K_2SO_4 with 0.5 M Na_2CO_3 .

Original Comment 2. *The nature of good stability of $\text{KF}\cdot\text{H}_2\text{O}_2$ structure has also been reported (<https://doi.org/10.1023/A:1013040717382>), which is understandable given the strong hydrogen bond between F and H-O. However, the author introduced a concept of “borrowed blurring effect” but did not clarify the origin and meaning, or explain the chemical similarity in their finding.*

Response: We sincerely thank the reviewer for his/her useful comment. Our response is as follows:

(1) Yes, there are indeed a few examples reporting the $\text{KF}\cdot\text{H}_2\text{O}_2$ structure (like Russ. J. Appl. Chem. 74, 907–911, (2001) and Russ. Chem. Bull. 42, 30–35, (1993)), but only focused on exploring its chemical stability or traditional synthesis procedure characteristic of high cost, complex, and rather low productivities. In contrast, here we have reported a novel electrochemical synthesis technique sourced from air,

enabling the green and efficient production of peroxosolvates. Notable, our synthetic procedure achieves a remarkable production rate up to 1.6 times of benchmark industrial anthraquinone process, which represents a significant advance for practical synthesis of peroxosolvates.

(2) Next, the blurring effect describes hydrogen-bond-induced charge delocalization. Analogous to Gaussian blurring in image processing averages pixel intensities to reduce local contrast, here we explain this effect as homogenizing electron distribution across H-O and O-O bonds within H₂O₂ molecules *via operando* hydrogen-bond networks (e.g., KF·H₂O₂ formation), thereby suppressing H₂O₂ decomposition. We have further clarified the concept of “borrowed blurring effect” as follows:

To further validate the stabilizing role of blurring effect on H₂O₂ molecule, we have calculated hydrogen-bond energies for isolated H₂O₂ and H₂O₂ in peroxosolvates (i.e., KF·H₂O₂, CO(NH₂)₂·H₂O₂ and Na₂CO₃·1.5H₂O₂, Supplementary Fig. 33)²⁷. Generally, the instability of H₂O₂ molecule originates from nonuniform charge distributions between different bonds, with a bonding energy difference of 1.07 eV between O-H and O-O bonds. Interestingly, both bond energies of O-H and O-O in peroxosolvate have been reduced, attributable to bond elongation induced by hydrogen bonding (i.e., O-H···N, O-H···O and O-H···F, Supplementary Fig. 34). The difference between the O-H and O-O bond energies in peroxosolvates has also been reduced to 0.90 eV for KF·H₂O₂, 0.91 eV for CO(NH₂)₂·H₂O₂ and 0.92 eV for Na₂CO₃·1.5H₂O₂, respectively (Supplementary Fig. 33d). Accordingly, we consider the hydrogen bonding has introduced blurring effect between O-H and O-O bonds, enabling high stability of H₂O₂ molecule. (Page 8 line 24, Page 9 line 1–10)

The blurring effect is not exclusive to KF. Our additional tests using NaF and K₂CO₃ as electrolytes were conducted in the same condition. For NaF electrolyte, the H₂O₂ Faradaic efficiencies (FEs) of ZIF-350 consistently exceeded 90% across the entire range of applied current densities, the values being 97.5%~90.8% at 50~350 mA cm⁻², respectively (Supplementary Fig. 38a). Even at the theoretical limiting

current density of air-to-H₂O₂ solidification ($\sim 350 \text{ mA cm}^{-2}$), the FE reaches 90.8%, together with the exceptional H₂O₂ yield rate of $29.62 \text{ mol g}_{\text{cat}}^{-1} \text{ h}^{-1}$. On the other hand, for K₂CO₃ electrolyte, the H₂O₂ Faradaic efficiencies (FEs) of ZIF-350 is 97.9%~87.8% at $50\sim 350 \text{ mA cm}^{-2}$ (Supplementary Fig. 38b), and reaching 87.8% and $28.64 \text{ mol g}_{\text{cat}}^{-1} \text{ h}^{-1}$ at $\sim 350 \text{ mA cm}^{-2}$. These results suggest that electrolytes capable of forming hydrogen bonds with H₂O₂ can also introduce blurring effect. (Page 9 line 22–24, Page 10 line 1–7)

Supplementary Figure 33. Bond energy of isolated H₂O₂ and H₂O₂ in peroxosolvates. a, Bond energy of O-H...N/O/F. **b**, Bond energy of O-H. **c**, Bond energy of O-O. **d**, Bond energy difference between O-H and O-O bonds.

Supplementary Figure 34. Bond length of isolated H_2O_2 and H_2O_2 in peroxosolvates. a, Bond length of O-H in H_2O_2 . b, Bond length of O-O in H_2O_2 . c, Schematic of stabilisation of H_2O_2 by blurring effect.

Supplementary Figure 38. The 2e-ORR performances of ZIF-350 in NaF and K_2CO_3 electrolytes. a, 0.5 M NaF. b, 1.0 M K_2CO_3 .

Original Comment 3. *To experimentally validated the stabilizing effect of $\text{KF} \cdot \text{H}_2\text{O}_2$, the author conducted Raman studies and claimed that the formation of strong hydrogen bond led to a shift in O-O peak from 874cm^{-1} (commercial) to 858cm^{-1} ($\text{KF} \cdot \text{H}_2\text{O}_2$). However, in fig3, the signal-to-noise ratio for the $\text{KF} \cdot \text{H}_2\text{O}_2$ sample in Fig. 3a is too low to distinguish the peak from background, which undermines the reliability of the interpretation of the Raman spectra.*

Response: Thanks for your kind reminding. We have provided the response as follows:

Generally, Raman spectra is a scattering signal with intrinsic weak intensity. When used to study the structural characteristics of a bulk material, obvious signal peaks can be seen. This is due to the structural crystal lattice that contributes to overall vibration of the bulk material, leading to enhanced scattered signals.

While in this work, the *operando* Raman only probes the signals of O-O stretching vibrations in H₂O₂ during catalytic processes. The scattered Raman signals are mainly focused on bond vibrations of adsorbed species (like *OOH in 2e-ORR) on catalyst surfaces, which are known to be very weak as comparison to structural crystal lattices in bulk materials. Consequently, the *operando* Raman signals for catalytic reactions is mostly very weak in the literatures (like ORR, NRR, CRR).^{J. Am. Chem. Soc. 142, 715-719 (2020); Angew. Chem. Int. Edit. 60, 20331-20341 (2021); ACS Nano 14, 11363-11372 (2020)} Our *operando* Raman signals in Figure 3a and Supplementary Figure 32 are comparable to the above literature.

To confirm the accuracy of the experimental results, we performed 10 repetitive *operando* Raman tests. We directly determined the positions and intensities of the peaks using Raman spectrometer software, and updated the *operando* Raman data with error bars. By comparing the Raman vibration of ZIF-350 at different electrolysis times, we conclude the formation of strong hydrogen bond leading to a shift in O-O peak from 874cm⁻¹ (commercial) to 858cm⁻¹ (KF·H₂O₂). Further, the intensity of *OOH peak increased consistently with elongated reaction duration, proving the validity of our *operando* synthesis condition.

Based on above discussions, we have made relevant changes in the updated manuscript as follows:

Supplementary Figure 32. Ten repetitive *operando* Raman tests for ZIF-350 in air under the same condition. a, 4 hrs. b, 2 hrs. c, 0 hr. d, The 858 cm⁻¹ peak intensity with error bar in *operando* Raman test. Ten measurements were conducted for each data point with the error bars corresponding to the standard deviation.

Supplementary Note 5:

Additional ten repetitive tests were performed for *operando* Raman under the same condition of Supplementary Figure 32. The weak Raman signal is due to the harsh *operando* test condition¹¹. In this work, the *operando* Raman only probes the signals of O-O stretching vibrations in H₂O₂ during catalytic processes. The scattered Raman signals are mainly focused on bond vibrations of adsorbed species (like *O₂ and *OOH in ORR) in electrolyte, which are known to be very weak as comparison to bulk materials. To confirm the accuracy of the experimental results, Raman spectrometer software is used to directly determine the peak positions and intensities. On this basis, the Raman information of H₂O₂ in KF are compared at different durations. By comparing the Raman vibration of ZIF-350 at different electrolysis durations^{12, 13}, we conclude the intensity of *OOH peak increased consistently with elongated reaction duration, proving the validity of our *operando* synthesis condition.

We would like to thank the Editor's invitation to revise the manuscript. We are also appreciated of the reviewer #3 for his/her positive recommendation and constructive comments to improve the quality of this work.

After receiving the reviewer's comments, we have spent several intensive weeks in order to answer the concerns raised by the astute reviewer. We believe we have shifted the work to the next level of proof and gained a much deeper understanding of the system. Attached please see the point-by-point response to the reviewer #3's comments. Thank you and best wishes,

Kind regards

The authors

List of changes

- (1) We have described in details the concept of blurring effects and modify the relevant sections in the main text. (**Response to Reviewer #3 Comment 1**; Page 5 line 7–13; Page 9 line 22–24; Page 10 line 1–7; Supplementary Figures 3 and Supplementary Figures 38)
- (2) We have modified the relevant descriptions of *operando* Raman and added the calculations of hydrogen bonding interactions in peroxosolvates. (**Response to Reviewer #3 Comment 2**; Page 8 line 17–20; Page 8 line 24; Page 9 line 1–10; Supplementary Figures 3; Supplementary Figures 33–34; Supplementary Figures 38; Reference 27)
- (3) We have conducted additional calculations for molecular dynamics (MD) and have provided corresponding content in the main text and supplementary information. (**Response to Reviewer #3 Comment 3**; Page 9 line 10–14; Supplementary Figure 35; Reference 28)
- (4) We have added the calculations of bond strengths of isolated H₂O₂ and H₂O₂ in peroxosolvates. (**Response to Reviewer #3 Comment 4**; Page 8 line 24; Page 9 line 1–10; Supplementary Figures 33–34; Reference 27)

Response to Reviewer #3

Original Comment: *This manuscript presents an experimental and computational study of the direct solid-state H₂O₂ electrosynthesis and storage in the form of peroxosolvates of KF·H₂O₂, CO(NH₂)₂·H₂O₂ and Na₂CO₃·1.5H₂O₂. The as-fabricated solid-state H₂O₂ features not only high gravimetric densities but also good stability for repeated loading/deloading and shelf life. DFT calculations indicate the charge distribution of H-O and O-O bonds of H₂O₂ in KF·H₂O₂ becomes homogenizing compared with that of H₂O₂ molecule. Overall, this manuscript is interesting, and the results are properly presented and discussed. However, parts of the discussion lack evidences and more calculations are needed to support their conclusions. I therefore recommend a major revision before considering publication.*

Original Comment 1. *I think the stable load/deloading of H₂O₂ with KF, etc. is related the regulation of the H-O and O-O bond strength in H₂O₂ through the formation of possible hydrogen bond between H₂O₂ and KF. The “blurring effect” is not strongly related to the manuscript and should be revised to a more straightforward statement, such as “the regulation of bond length by the formation of possible hydrogen bond between H₂O₂ and F, N and O”. I suggest the author revise the corresponding part of the manuscript.*

Response: Thanks for the kind reminding. We have added the recommended statement into updated manuscript (Page 5 line 7–13), and further elaborated the relationship between blurring effect and hydrogen bond as follows:

1) Generally, the blurring effect describes hydrogen-bond-induced charge delocalization. Analogous to Gaussian blurring in image processing averages pixel intensities to reduce local contrast, we explain this effect as homogenizing electron distribution across H-O and O-O bonds within H₂O₂ molecules *via operando* hydrogen-bond networks (e.g., KF·H₂O₂ formation), thereby suppressing H₂O₂ decomposition.

2) Theoretically, our DFT calculations reveal that *operando* hydrogen bonds in KF·H₂O₂ significantly alter electronic structures (Fig. 3b). The KF*H₂O₂ complex

exhibits stronger adsorption energy (-0.90 eV) compared to $\text{KF}\cdot\text{H}_2\text{O}$ (-0.50 eV), confirming enhanced stability. Hydrogen bonding induces a charge transfer of 0.67 e from H-O bonds and 0.22 e from O-O bonds, homogenizing electron density distributions.

3) Experimentally, we have added FTIR test to explain the hydrogen bonding between a hydrogen atom and F/N/O. As shown in Supplementary Figure 3a, K-F vibration at 603 cm^{-1} and O-H at 3292 cm^{-1} . While in $\text{KF}\cdot\text{H}_2\text{O}_2$, both vibrations have changed significantly, i.e., 571 cm^{-1} for K-F and 3036 cm^{-1} for O-H vibration. According to the response to Comment 4, the peak shift is originated to the formation of O-H \cdots F bonds. The same phenomenon has been observed for $\text{CO}(\text{NH}_2)_2\cdot\text{H}_2\text{O}_2$ and $\text{Na}_2\text{CO}_3\cdot 1.5\text{H}_2\text{O}_2$.

4) Further, the blurring effect is not exclusive to KF. Our additional tests using NaF and K_2CO_3 as electrolytes was conducted in the same condition. For NaF electrolyte, the H_2O_2 Faradaic efficiencies (FEs) of ZIF-350 consistently exceeded 90% across the entire range of applied current densities, the values being 97.5%~90.8% at 50~350 mA cm^{-2} , respectively (Supplementary Fig. 38a). Even at the theoretical limiting current density of air-to- H_2O_2 solidification ($\sim 350\text{ mA cm}^{-2}$), the FE reaches 90.8%, together with the exceptional H_2O_2 yield rate of $29.62\text{ mol g}_{\text{cat}}^{-1}\text{ h}^{-1}$. On the other hand, for K_2CO_3 electrolyte, the H_2O_2 Faradaic efficiencies (FEs) of ZIF-350 is 97.9%~87.8% at 50~350 mA cm^{-2} (Supplementary Fig. 38b), and reaching 87.8% and $28.64\text{ mol g}_{\text{cat}}^{-1}\text{ h}^{-1}$ at $\sim 350\text{ mA cm}^{-2}$. This suggests that those electrolytes capable of forming hydrogen bonds with H_2O_2 can also produce blurring effect.

5) Overall, the blurring effect requires additives capable of forming hydrogen bonds (e.g., O-H \cdots F, O-H \cdots O, O-H \cdots N) with H_2O_2 , where the electronegative atom (F/O/N) homogenizes charge distribution across O-O and O-H bonds. As a consequence, whenever H_2O_2 is generated at the local environment of electrode-electrolyte interface, it interacts with the additive, which is crucial for the stabilization.

Based on above discussions, we have modified relevant descriptions in the main

text:

Here we extend this concept to electrochemistry and describe a novel stabilization mechanism, where the blurring effect arises from charge redistribution mediated by *operando* hydrogen bonds between electrochemically generated H_2O_2 and host compounds (e.g., KF). This concept then homogenizes electron density across H-O and O-O bonds, blurring their localized charge polarization to prevent bond cleavage, and promoting solid-state H_2O_2 electrosynthesis approaching theoretical limit efficiency (93.3% at the limit current of $\sim 350 \text{ mA cm}^{-2}$). (Page 5 line 7–13)

Supplementary Figure 3. FTIR spectra of peroxosolvates. a, KF and $\text{KF}\cdot\text{H}_2\text{O}_2$. **b,** $\text{CO}(\text{NH}_2)_2$ and $\text{CO}(\text{NH}_2)_2\cdot\text{H}_2\text{O}_2$. **c,** Na_2CO_3 and $\text{Na}_2\text{CO}_3\cdot 1.5\text{H}_2\text{O}_2$. Generally, KF alone only show typically K-F vibration at 603 cm^{-1} and O-H at 3292 cm^{-1} . While for $\text{KF}\cdot\text{H}_2\text{O}_2$, both vibrations have changed significantly, *i.e.*, 571 cm^{-1} for K-F and 3036 cm^{-1} for O-H vibration. The peak shift might be originated to the formation of O-H \cdots F bonds. The same phenomenon has been observed for $\text{CO}(\text{NH}_2)_2\cdot\text{H}_2\text{O}_2$ and $\text{Na}_2\text{CO}_3\cdot 1.5\text{H}_2\text{O}_2$ counterparts.

The blurring effect is not exclusive to KF. Our additional tests using NaF and K_2CO_3 as electrolytes were conducted in the same condition. For NaF electrolyte, the H_2O_2 Faradaic efficiencies (FEs) of ZIF-350 consistently exceeded 90% across the

entire range of applied current densities, the values being 97.5%~90.8% at 50~350 mA cm⁻², respectively (Supplementary Fig. 38a). Even at the theoretical limiting current density of air-to-H₂O₂ solidification (~350 mA cm⁻²), the FE reaches 90.8%, together with the exceptional H₂O₂ yield rate of 29.62 mol g_{cat}⁻¹ h⁻¹. On the other hand, for K₂CO₃ electrolyte, the H₂O₂ Faradaic efficiencies (FEs) of ZIF-350 is 97.9%~87.8% at 50~350 mA cm⁻² (Supplementary Fig. 38b), and reaching 87.8% and 28.64 mol g_{cat}⁻¹ h⁻¹ at ~350 mA cm⁻². These results suggest that electrolytes capable of forming hydrogen bonds with H₂O₂ can also introduce blurring effect. (Page 9 line 22–24, Page 10 line 1–7)

Supplementary Figure 38. The 2e-ORR performances of ZIF-350 in NaF and K₂CO₃ electrolytes. a, 0.5 M NaF. b, 1.0 M K₂CO₃.

Original Comment 2. *I think the statement of “this band is shifted by 15 cm⁻¹ when compared to that of commercial H₂O₂ (873 cm⁻¹), which is due to the operando hydrogen bonds between the electrochemically produced H₂O₂ and KF in the electrolyte” is too strong and should be revised. The formation of operando hydrogen bonds between H₂O₂ and KF in the electrolyte lacks direct evidence, and is just speculated from the operando Raman, according to the vibration frequency of O-O bond. H position generally is very difficult to be determined by XRD. Raman spectroscopy measures the vibrational mode but is also insufficient to detect the H position. Instead, neutron diffraction could determine the H position and related H bond, see *Inorganics* 2022, 10, 132. Here, I only suggest the author calculate the bond strength of the H bond (O-H···F, O-H···O and O-H···N), and the O-H bonds of*

KF·H₂O₂, CO(NH₂)₂·H₂O₂ and Na₂CO₃·1.5H₂O₂, and compare them with the O-H bond strength of isolated H₂O₂ molecule (a very related and nice paper could be referred and cited, Inorganics 2022, 10, 132), to provide an indirect supporting of this speculation.

Response: We are appreciated of the reviewer for his/her constructive comment. Accordingly, we have calculated the bond strength of relevant H bonds, cited recommended literature (Inorganics 2022, 10, 132), and given the response as follows:

1) The relevant sentences have been revised as “Notably, the Raman band in the spectrum is shifted by 15 cm⁻¹ compared to commercial H₂O₂ (873 cm⁻¹). This shift may be attributed to the interaction between KF and H₂O₂ produced electrochemically in the electrolyte, and it is speculated that *operando* hydrogen bonds may be formed in this process.” (Page 8 line 17–20)

2) To further validate the stabilizing role of blurring effect on H₂O₂ molecule, we have calculated hydrogen-bond energies for isolated H₂O₂ and H₂O₂ in peroxosolvates (*i.e.*, KF·H₂O₂, CO(NH₂)₂·H₂O₂ and Na₂CO₃·1.5H₂O₂, Supplementary Fig. 33)²⁷. Generally, the instability of H₂O₂ molecule originates from nonuniform charge distributions between different bonds, with a bonding energy difference of 1.07 eV between O-H and O-O bonds. Interestingly, both bond energies of O-H and O-O in peroxosolvate have been reduced, attributable to bond elongation induced by hydrogen bonding (*i.e.*, O-H···N, O-H···O and O-H···F, Supplementary Fig. 34). The difference between the O-H and O-O bond energies in peroxosolvates has also been reduced to 0.90 eV for KF·H₂O₂, 0.91 eV for CO(NH₂)₂·H₂O₂ and 0.92 eV for Na₂CO₃·1.5H₂O₂, respectively (Supplementary Fig. 33d). Accordingly, we consider the hydrogen bonding has introduced blurring effect between O-H and O-O bonds, enabling high stability of H₂O₂ molecule. (Page 8 line 24, Page 9 line 1–10)

Supplementary Figure 33. Bond energy of isolated H₂O₂ and H₂O₂ in peroxosolvates. a, Bond energy of O-H...N/O/F. b, Bond energy of O-H. c, Bond energy of O-O. d, Bond energy difference between O-H and O-O bonds.

Supplementary Figure 34. Bond length of isolated H₂O₂ and H₂O₂ in peroxosolvates. a, Bond length of O-H in H₂O₂. b, Bond length of O-O in H₂O₂. c, Schematic of stabilisation of H₂O₂ by blurring effect.

Ref 27. Benz S., et al. The crystal structure of carbonic acid. *Inorganics* **10**, 132 (2022)

Original Comment 3. *Given that theory and multi-scale modeling and simulation, as supplements to experimental efforts, can help greatly to close some of the current experimental and technological gaps, as well as predict path-independent properties and help to fundamentally understand path-independent performance in multiple spatial and temporal scales, some relevant references are necessary to review to enhance the readability of the manuscript: e.g., Chinese Physics B 25, 018212 (2016).*

Response: We are appreciated of the reviewer for his/her useful comment. We have reviewed and cited the recommended literature (Ref 28. Chinese Physics B 25, 018212 (2016)). Further, we have conducted molecular dynamics (MD) simulations to understand some of the experimental and technological gaps as follows:

Molecular dynamics (MD) simulations provide further insight into the blurring effect (Supplementary Fig. 35)²⁸. It demonstrates that during energy minimization, H₂O₂ molecules undergo directed migration towards KF species. This spontaneous reconfiguration, driven by favorable electrostatic interactions and hydrogen bonding, leads to a significant reduction in the total system energy. (Page 9 line 10–14)

Supplementary Figure 35. MD simulations of the structures and dynamics of KF and H₂O₂. a, Kinetic energy. b, Thermodynamic energy. c, The optimized KF and

H₂O₂ model.

Ref 28. Shi S. Q., *et al.* Multi-scale computation methods: their applications in lithium-ion battery research and development. *Chinese Phys. B* **25**, 018212 (2016).

Original Comment 4. *The calculation of the bond strengths of H-O and O-O of H₂O₂ molecules in KF·H₂O₂, CO(NH₂)₂·H₂O₂, Na₂CO₃·1.5H₂O₂ and isolated H₂O₂ molecular is indispensable, because it could give a direct and quantitative data (instead of the very weak connection between the bond strength and charge transfer) to support their conclusions “inhibiting the break of these bondings inside H₂O₂ molecules”.*

Response: Thanks for the kind reminding. We have calculated the bond strengths of H-O and O-O of H₂O₂ molecules in KF·H₂O₂, CO(NH₂)₂·H₂O₂, Na₂CO₃·1.5H₂O₂ and isolated H₂O₂ molecule. We have given the discussions as follows:

To further validate the stabilizing role of blurring effect on H₂O₂ molecule, we have calculated hydrogen-bond energies for isolated H₂O₂ and H₂O₂ in peroxosolvates (*i.e.*, KF·H₂O₂, CO(NH₂)₂·H₂O₂ and Na₂CO₃·1.5H₂O₂, Supplementary Fig. 33)²⁷. Generally, the instability of H₂O₂ molecule originates from nonuniform charge distributions between different bonds, with a bonding energy difference of 1.07 eV between O-H and O-O bonds. Interestingly, both bond energies of O-H and O-O in peroxosolvate have been reduced, attributable to bond elongation induced by hydrogen bonding (*i.e.*, O-H···N, O-H···O and O-H···F, Supplementary Fig. 34). The difference between the O-H and O-O bond energies in peroxosolvates has also been reduced to 0.90 eV for KF·H₂O₂, 0.91 eV for CO(NH₂)₂·H₂O₂ and 0.92 eV for Na₂CO₃·1.5H₂O₂, respectively (Supplementary Fig. 33d). Accordingly, we consider the hydrogen bonding has introduced blurring effect between O-H and O-O bonds, enabling high stability of H₂O₂ molecule. (Page 8 line 24, Page 9 line 1–10)

Supplementary Figure 33. Bond energy of isolated H_2O_2 and H_2O_2 in peroxosolvates. a, Bond energy of O-H...N/O/F. b, Bond energy of O-H. c, Bond energy of O-O. d, Bond energy difference between O-H and O-O bonds.

Supplementary Figure 34. Bond length of isolated H_2O_2 and H_2O_2 in peroxosolvates. a, Bond length of O-H in H_2O_2 . b, Bond length of O-O in H_2O_2 . c, Schematic of stabilisation of H_2O_2 by blurring effect.

Response to Reviewer #2

Original Comment: *The reviewer appreciated the author's effort in providing a clearer discussion on the process and mechanism. However, several concerns remained in the presentation of performance and mechanism.*

Original Comment 1. *The proposed and described novel H₂O₂ stabilization mechanism: "blurring effect" is conceptually the same as forming a stronger hydrogen bond network that is widely applied in the field of stabilizing H₂O₂. It is recommended that the author present the conceptual advancement beyond the previous knowledge.*

Response: Thanks for your kind reminding. Following the reviewer's kind suggestion, we would like to propose a new concept of "electronic structure blurring", and give more details as follows:

Traditionally, "blurring effect" is a well-known phenomenon in image processing research field. It is a parasitic process of removing noise and artifacts during image optimization, which exhibits negative role of reducing image quality. There are seldom studies of "blurring effect" in chemistry/catalysis.

In the present work, we make a conceptual advancement by blending "blurring effect" with chemistry/catalysis, forming a new concept of "electronic structure blurring". Our study starts by trying to find a mechanism to understand an unexpected phenomenon, *i.e.*, H₂O₂ stability significantly enhanced in the presence of KF, which greatly promote the electrocatalytic activities. We give the explanation from perspective of electron structure. Our DFT calculations shows the electronic cloud distribution of H₂O₂ and KF homogenizing across F···H-O and O-O bonds.

More specifically, DFT calculation confirm the "electronic structure blurring" that drives charge transfer of 0.67 e for H-O bond and 0.22 e for O-O bond, thereby homogenizing the charge distribution across these bonds. This can reduce the difference in bond energies (from 1.07 eV in pure H₂O₂ to 0.90 eV in KF·H₂O₂), thereby inhibiting bond cleavage—the root cause of H₂O₂ decomposition.

Different from previous studies of explaining the stabilizing H_2O_2 by hydrogen bonds from the perspective of macroscopic physical confinement (*e.g.*, restricting molecular diffusion or aggregation through hydrogen bond interactions). In the present work, "electronic structure blurring" describes a more intrinsic, atomistic-level modulation for inhibiting H_2O_2 decomposition.

Based on above discussions, we have revised all relevant sections by using the new concept of "electronic structure blurring". (Title of manuscript; Page 2 line 5, 10 and 14; Page 5 line 7–8; Page 8 line 22; Page 9 line 6–8 and 19–20; Page 10 line 4; Page 11 line 5; Page 13 line 3–4; Figure 3)

Figure 3 | Mechanistic investigation. **a**, The *operando* Raman spectra for air-to- H_2O_2 conversion. **b**, Adsorption energy of KF with H_2O_2 and H_2O (inset shows the charge transport inside $\text{KF}^*\text{H}_2\text{O}_2$). **c**, H_2O_2 yield rates in different KF concentrations in flow cells. **d**, Free-energy diagrams for 2 and 4e-ORR pathways. **e**, The theoretical volcano curve for 2e-ORR. **f**, Kinetic barriers for $^*\text{O}_2 \rightarrow ^*\text{OOH}$ via Eley-Rideal (E-R) and Langmuir-Hinshelwood (L-H) mechanisms. **g**, Schematic air-to- H_2O_2 solidification pathway.

Original Comment 2. *The yield of H₂O₂ and multiple related performance parameters were evaluated by quantifying the H₂O₂ concentration in the electrolyte. While the major technical advancement was given to the formation of solid-state KF·H₂O₂, the H₂O₂ yield should also be analyzed in the final solid form to provide a complete analysis in case of potential H₂O₂ loss during the solidification.*

Response: Thank you for your valuable suggestion. Yes, as pointed out by the astute reviewer, there might be potential H₂O₂ loss during solidification, which causes data errors in determining the productivity of KF·H₂O₂. Consequently, we have added two comparison experiments to determine the potential H₂O₂ loss during the solidification as follows:

(1) We have prepared the following solutions: *i*) H₂O₂ in KF electrolyte (500 mg L⁻¹, the concentration in our electrosynthesis system); *ii*) solid KF·H₂O₂ in KF electrolyte (equivalent amount).

As shown in Supplementary Fig. 24a, the UV-Vis spectra of both solutions were recorded using the KI-based colorimetric method (351 nm). The results showed identical absorption peaks with seldom peak intensity difference (3.05%). This phenomenon indicates that UV-vis quantification method can accurately reflect the KF·H₂O₂ concentration by using H₂O₂ in KF electrolyte.

(2) Further, we have electrochemically synthesized KF·H₂O₂ at the current density of 350 mA cm⁻². The concentration of as-produced KF·H₂O₂ in the electrolyte was determined by UV-Vis as 12.31 mmol L⁻¹.

Next, the KF·H₂O₂-containing electrolyte was evaporated and dried under vacuum (40 °C) to obtain solid KF·H₂O₂. The mass of solid-state KF·H₂O₂ was determined by UV-Vis to be 12.06 mmol L⁻¹. The difference of KF·H₂O₂ before and after solidification was minor (2.03%, Supplementary Fig. 24b), indicating seldom loss of H₂O₂ during solidification and drying process.

Based on above discussions, we conclude that all electrochemical data is valid for use in our study. The following sections have been added into updated manuscript:

Supplementary Figure 24. Comparison experiments to determine the potential H₂O₂ loss during the solidification. **a**, UV-vis spectra of KF electrolyte with commercial H₂O₂ (500 mg L⁻¹) and KF electrolyte with an equivalent amount of solid KF·H₂O₂. **b**, UV-vis spectra of KF·H₂O₂ under electrolysis at 350 mA cm⁻² before and after solidification.

Supplementary Note 5:

There might be potential H₂O₂ loss during the solidification, which causes data errors in determining the productivity of KF·H₂O₂. Consequently, we have added two comparison experiments to determine the potential H₂O₂ loss during the solidification as follows:

(1) We have prepared the following solutions: *i*) H₂O₂ in KF electrolyte (500 mg L⁻¹, the concentration in our electrosynthesis system); *ii*) solid KF·H₂O₂ in KF electrolyte (equivalent amount). As shown in Supplementary Fig. 24a, the UV-Vis spectra of both solutions were recorded using the KI-based colorimetric method (351 nm). The results showed identical absorption peaks with seldom peak intensity difference (3.05%).

This phenomenon indicates that UV-Vis quantification method can accurately reflect the $\text{KF}\cdot\text{H}_2\text{O}_2$ concentration by using H_2O_2 in KF electrolyte.

(2) Further, we have electrochemically synthesized $\text{KF}\cdot\text{H}_2\text{O}_2$ at the current density of 350 mA cm^{-2} . The concentration of as-produced $\text{KF}\cdot\text{H}_2\text{O}_2$ in the electrolyte was determined by UV-Vis as $12.31 \text{ mmol L}^{-1}$. Next, the $\text{KF}\cdot\text{H}_2\text{O}_2$ -containing electrolyte was evaporated and dried under vacuum ($40 \text{ }^\circ\text{C}$) to obtain solid $\text{KF}\cdot\text{H}_2\text{O}_2$. The mass of solid-state $\text{KF}\cdot\text{H}_2\text{O}_2$ was determined by UV-Vis to be $12.06 \text{ mmol L}^{-1}$. The difference of $\text{KF}\cdot\text{H}_2\text{O}_2$ before and after solidification was minor (2.03%, Supplementary Fig. 24b), indicating seldom loss of H_2O_2 during solidification and drying process.

Based on above discussions, we conclude that all of the electrochemical data is valid for use in our study.

Original Comment 3. *Several important testing parameters were missing from the discussion and the method section. For example, the H_2O_2 concentration in the electrolyte before evaporation and solidification, and the air flow rate used at each current density for the flow cell tests. The fabrication method of the catalyst-coated carbon fiber paper and how the catalyst loading was determined. These parameters were essential in evaluating normalized performance in electrocatalysis.*

Response: Thanks for your kind reminding. We have added these testing parameters into updated manuscript as follows:

(1) The H_2O_2 concentration in the electrolyte before evaporation and solidification is $65.7\text{--}431.1 \text{ mg L}^{-1}$ ($50\text{--}350 \text{ mA cm}^{-2}$) and $173.6\text{--}1104.4 \text{ mg L}^{-1}$ ($0.5\text{--}3.5 \text{ A}$). (Supplementary Figures 23 and 47)

(2) The air flow rate used at each current density for the flow cell tests is 100 sccm .

(Page 15 line 1)

(3) The fabrication method of the catalyst-coated carbon fiber paper is as follows:

In detail, 5 mg of catalysts and 50 μL of Nafion solution (5 wt%) were dispersed in 950 μL of isopropanol by ultrasonication for 1 hr to form a uniform catalyst ink. The ink was then drop-cast onto carbon fiber paper (28BC) as a cathode using a micropipette. The catalyst loading of 0.2 mg cm^{-2} was determined by weighing the mass of carbon fiber paper before and after spraying, which was dried under ambient conditions to form gas diffusion electrolyte (Supplementary Table 10). (Page 14 line 15–18)

(4) The catalyst loading was determined by the following method:

Supplementary Table 10. Average mass loadings of ZIF-350 catalysts. Actual test area is 1 cm^2 .

m (carbon fiber paper)/ mg	m (ZIF-350 on carbon fiber paper)/ mg	Area/cm^{-2}	Mass loading/mg cm^{-2}
33.47	34.04	3.0	0.190
35.12	35.74	3.0	0.207
36.89	37.43	3.0	0.180
34.25	34.94	3.0	0.230
37.56	38.14	3.0	0.193
32.15	32.80	3.0	0.217
35.78	36.38	3.0	0.200

33.91	34.46	3.0	0.183
36.23	36.89	3.0	0.220
34.67	35.26	3.0	0.197
average			0.202

Supplementary Note 7:

The catalyst loading was determined by the following method:

$$\text{Mass loading} = (m_2 - m_1)/A$$

where m_1 is the mass of carbon fiber paper, m_2 is the mass of ZIF-350 on carbon fiber paper, A is the geometric of carbon fiber paper.

To obtain a more data, we repeated the material synthesis for ten times, and then calculate the average mass loading for ZIF-350.

Original Comment 4. *An impressive production rate was presented in Figure 4d (>250 mol/g/h). However, the reviewer had difficulties in the calculation process since the scaling up of device area should not benefit the mass-specific production rate (total catalyst loading should also be scaled up). Therefore, using 3.5A, 0.2mg/cm², 89.6%FE, 16cm² should give a production rate of 18.28 instead of 292.35 mol/g/h, which is 16x of this value.*

Response: Thanks for your kind reminding. Accordingly, we have rechecked the data, and re-calculated the production rate by using both catalyst mass per electrode area (0.2 mg cm⁻²) and total catalyst mass in cell stack (0.2 mg cm⁻² × 16 cm² = 3.2 mg) as follows:

$$\text{Yield rate} = \frac{C_{H_2O_2} \times V_{\text{electrolyte}}}{m_{\text{cat}} \times t}$$

$$\text{Total yield rate} = \frac{C_{H_2O_2} \times V_{\text{electrolyte}}}{t}$$

Where $C_{H_2O_2}$ is H_2O_2 concentration (mg L^{-1}), $V_{\text{electrolyte}}$ is the volume of cathodic electrolyte (L), F is the Faradaic constant (96485 C mol^{-1}), t is reaction duration and m_{cat} is catalyst mass loading (mg cm^{-2}).

Indeed, the production rate per electrode mass should be $18.25 \text{ mol g}_{\text{cat}}^{-1} \text{ h}^{-1}$, and total yield rate should be 58.47 mol h^{-1} . We have made the mistake by not considering the scaled up of catalyst loadings in the original manuscript. Now, we have made corrections/revisions in the relevant sections as follows:

Figure 4 | Scale-up production. **a**, Schematic configuration of our developed tandem/parallel prototype device. **b**, The optical picture of the prototype device. **c**, The H_2O_2 yield rates under ampere-level currents. **d**, The comparison of yield rates with the state-of-the-art literature for air-to- H_2O_2 conversion (please also see Supplementary Table 9). **e**, XRD patterns of electro-synthesized $KF \cdot H_2O_2$ as comparison to commercial counterpart. **f**, The stability test at 1 A for 50 hrs. The red

arrows represent electrolyte refresh. **g**, Potential application of $\text{KF}\cdot\text{H}_2\text{O}_2$ for the degradation of methylene blue pollutants.

Supplementary Figure 48. The 2e-ORR activity of ZIF-350 catalyst tested in 0.5 M K_2SO_4 under the same test condition.

Supplementary Table 9. Comparison of the electrochemical air-to- H_2O_2 conversion activities of ZIF-350 with state-of-the-art catalysts.

Catalyst	Electrolyte	Current (A)	FE (%)	Yield (mol h ⁻¹)	Reference ^{8, 14-21}
ZIF-350	1.0 M KF	3.5	89.62	58.47	This work
ER-ZnO	0.6 M K_2SO_4	0.3	89.3	4.88	Nat Commun. 15, 4157 (2024)
N, S-TCNTs	1.0 M KOH	0.35	93.0	15.19	Adv. Mater. 2023, 35, 2303905
Co HSACs	0.5 M KOH	0.3	90	5	Nat Commun. 14, 1426 (2023)
In SAs/NSBC	0.1 M KOH	0.63	77.34	9.09	Angew. Chem. Int. Ed.

					2022, 61 , e202117347
N, O-CNTs	1.0 M KOH	0.048	95	2.65	Adv. Sci. 2022, 9, 2201421
PD/N-C	0.1 M HClO ₄	0.1	89	10.74	J. Am. Chem. Soc. 2023, 145, 11589–11598
CoSP/MWCNTs	0.05 M Na ₂ SO ₄	0.8	60	6.36	Chem. Eng. J. 2020, 379, 122417
Co-NC/Mxenes	0.5 M H ₂ SO ₄	0.5	90	3.02	Appl. Catal. B Environ. 2022, 317, 121737
CoPc/CNT	0.1 M H ₂ SO ₄	0.48	/	17.81	Chin. J. Catal. 2022, 43 (5), 1238–1246
FCB	SEC	0.8	61.5	42.32	Science 366, 226-231 (2019)

Page 11 line 22–25, Page 12 line 1–3:

Analogously, the corresponding H₂O₂ yield rates are 9.19, 17.61, 26.67 and 35.42 mol h⁻¹ at applied currents from 0.5, 1.0, 1.5 and 2.0 A, respectively. Even by further elevating the current to 3.5 A, the prototype device can still maintain high Faradaic efficiency of 89.6% and H₂O₂ yield rate of 58.47 mol h⁻¹ with the KF electrolyte. For reference, we also analyzed the standard K₂SO₄ electrolyte under the same conditions, and significant worse behavior could be noted (Faradaic efficiency of 47.27% and yield rate of 30.84 mol h⁻¹, Supplementary Fig. 48).

Borrowed blurring effect-mediated solid-state H₂O₂ electrosynthesis with productivity exceeding benchmark industrial route

This paper presents a solid-state electrosynthesis approach for H₂O₂ production, demonstrating a productivity that exceeds that of the conventional industrial process. Traditional H₂O₂ synthesis requires strategies to suppress its decomposition; in this study, the authors introduce peroxosolvates to mitigate H₂O₂ degradation, achieving a productivity of 0.943 mol L⁻¹ h⁻¹, which surpasses that of the benchmark anthraquinone method. The work is significant as it proposes a novel electrochemical system design for efficient peroxide production. I recommend this manuscript for publication with minor revisions. Specific comments are as follows:

Comments

1. The overall image quality is not satisfactory. It is necessary to enhance the resolution of the figures to improve readability.
2. In many cases, the figure descriptions are either missing, insufficient, or inaccurate

For example:

- There is no description provided for Figures 1a and 1b.
- Additional and more specific explanations are needed for the supplementary data.
- In Figure 1g, it is unclear why the storage capacity improves with increasing cycle number; this requires further clarification.
- The purpose of Figure 1f is not clearly explained.
- For Figures 1c and 1d, the schemes are not clearly described, making it difficult to understand what they represent.
- The explanation for Figure/Table 1 is insufficient.
- While the storage capacity appears to improve with an increasing number of cycles, this trend is not addressed in the text and should be discussed.
- There is no explanation provided for Figures 21 through 23.
- In Figure 40, the content of the illustration seems inconsistent with its description.

Overall, many parts of the figures and their corresponding explanations require revision and clarification.

3. Additional proofreading is necessary to correct typographical errors and verify the accuracy of the content. For example, in line 71, it should be confirmed whether "H₂O"

is the correct term in the phrase "direct air-to-diluted H₂O (<10 wt%)". Also, in line 86, it is unclear whether the reference to "Fig. 1c and d" is accurate. A more thorough review of such details is recommended.

4. The term "blurring effect" needs to be explained in more detail and with greater clarity regarding its meaning and implications. If possible, including a simulation that calculates and compares the bond dissociation energies of H–H and O–H would further strengthen the explanation.
5. In line 104, it is unclear why H is considered to be bonded with FNO. When explaining this using FTIR data, it would be helpful to include the specific peak assignment that supports this interpretation.
6. In Figure 2f, the experimental data for the 10% condition shows a higher current density; however, the corresponding yield appears to be lower. This discrepancy should be addressed or clarified.
7. In line 149, the Raman peak at 0 hours is not clearly visible. Similarly, the peak at 4 hours mentioned in line 151 also appears to be indistinct and should be re-examined.
8. The common characteristics of the additives used in this study—such as KF, Na₂CO₃, and urea—are not clearly explained. It would be helpful to clarify whether the blurring effect would still occur if alternative salts such as NaF were used instead of KF, or K₂CO₃ instead of Na₂CO₃. Furthermore, the specific conditions under which the blurring effect occurs should be described in more detail, ideally in relation to the physicochemical properties of the additives.
9. In Figure 37, it is unclear whether the illustrations of the Er and lh models are accurate. This should be carefully reviewed for correctness.